

# Ocean acidification in the North Atlantic: controlling mechanisms

Maribel I. García-Ibáñez[1], Patricia Zunino[2], Friederike Fröb[3], Lidia I. Carracedo[4], Aida F. Ríos[†], Herlé Mercier[5], Are Olsen[3], Fiz F. Pérez[1]

[1]Instituto de Investigaciones Marinas, IIM-CSIC, Vigo, E36208, Spain.
[2]Ifremer, Laboratoire de Physique des Océans, UMR 6523 CNRS/Ifremer/IRD/UBO, Ifremer Centre de Brest, Plouzané, CS 10070, France.
[3]Geophysical Institute, University of Bergen and Bjerknes Centre for Climate Research, Bergen, N5007, Norway.
[4]Faculty of Marine Sciences, University of Vigo, Vigo, E36200, Spain.
[5]CNRS, Laboratoire de Physique des Océans, UMR 6523 CNRS/Ifremer/IRD/UBO, Ifremer Centre de Brest, Plouzané, CS 10070, France.
[†]Deceased.

*Correspondence to*: Maribel I. García-Ibáñez (maribelgarcia@iim.csic.es)

**Abstract.** Repeated hydrographic sections provide critically needed data on, and understanding of, changes in basin-wide ocean $CO_2$ chemistry over multi-decadal timescales. Here, high-quality measurements collected at thirteen cruises carried out along the same track between 1981 and 2015, have been used to determine long-term changes in ocean $CO_2$ chemistry and ocean acidification in the Irminger and Iceland basins of the North Atlantic Ocean. Trends were determined for each of the main water masses present and are discussed in the context of the basin-wide circulation. The pH has decreased in all water masses of the Irminger and Iceland basins over the past 34 years, with the greatest changes in surface and intermediate waters (between $-0.0008 \pm 0.0001$ pH units·yr$^{-1}$ and $-0.0013 \pm 0.0001$ pH units·yr$^{-1}$). In order to disentangle the drivers of the pH changes, we decomposed the trends into their principal drivers: changes in temperature, salinity, total alkalinity ($A_T$) and total dissolved inorganic carbon (both its natural and anthropogenic components). The increase of anthropogenic $CO_2$ ($C_{ant}$) was identified as the main agent of the pH decline, partially offset by $A_T$ increases. The acidification of intermediate waters caused by $C_{ant}$ uptake has been reinforced by the aging of the water masses over the period of our analysis. The pH decrease of the deep overflow waters of the Irminger basin was similar to that observed in the upper ocean, and was mainly linked to the $C_{ant}$ increase, thus reflecting the recent contact of these deep waters with the atmosphere.

**Keywords.** Ocean acidification; $C_{ant}$; water masses; Subpolar Gyre.

## 1 INTRODUCTION

The oceanic uptake of a fraction of the anthropogenic $CO_2$ (i.e., $C_{ant}$; $CO_2$ released from humankind's industrial and agricultural activities) has resulted in long-term changes in ocean $CO_2$ chemistry, commonly referred to as ocean acidification, OA (e.g., Caldeira and Wickett, 2003, 2005; Raven et al., 2005; Doney et al., 2009; Feely et al., 2009). The changes in the ocean $CO_2$ chemistry result in declining pH and reduced saturation states for $CaCO_3$ minerals (e.g., Bates et al., 2014). The average pH ($-\log_{10}[H^+]$) of ocean surface waters has decreased by about 0.1 pH units since the beginning of the industrial revolution (1750), and based on model projections we expect an additional drop of 0.1–0.4 by the end of this century, even under conservative $CO_2$ emission scenarios (Caldeira and Wickett, 2005; Orr, 2011; Ciais et al., 2013). The rate of change in pH is at



least a hundred times faster than at any time since the last Ice Age (Feely et al., 2004; Raven et al., 2005), clearly
outpacing natural processes in ocean chemistry that have occurred in the past due to geological processes (Raven
et al., 2005). These changes in ocean $CO_2$ chemistry will most likely have adverse effects on organisms,
particularly calcifying ones, on ecosystems (e.g., Langdon et al., 2000; Riebesell et al., 2000; Pörtner et al.,
2004; Orr et al., 2005; Doney et al., 2009; Gattuso et al., 2014) and on major marine biogeochemical cycles (e.g.,
Gehlen et al., 2011; Matear and Lenton, 2014).
The global ocean has absorbed ~30% of the $C_{ant}$ emitted to the atmosphere between 1750 and the present
(Sabine et al., 2004; Khatiwala et al., 2013; DeVries, 2014; Le Quéré et al., 2015). This $C_{ant}$ is not evenly
distributed throughout the oceans (Sabine et al., 2004), but enters the interior ocean preferentially in regions of
deep convective overturn and subduction (Maier-Reimer and Hasselmann, 1987; Sarmiento et al., 1992; Lazier
et al., 2002). This explains why the Meridional Overturning Circulation (MOC) makes the North Atlantic Ocean
one of the most important $C_{ant}$ sinks of the global ocean, storing 25% of the global oceanic $C_{ant}$ (Sabine et al.,
2004; Khatiwala et al., 2013) despite being only 11% of the global ocean volume (Eakins and Sharman, 2010).
The MOC transports $C_{ant}$-laden surface waters from the Equator to the northern North Atlantic Ocean (e.g.,
Wallace, 2001; Anderson and Olsen, 2002; Álvarez et al., 2003; Olsen et al., 2006; Quay et al., 2007; Zunino et
al., 2015b), where deep water formation provides a pathway for $C_{ant}$ into the interior ocean (Lazier et al., 2002;
Pérez et al., 2008; Steinfeldt et al., 2009; Pérez et al., 2013). As regions close to deep water formation areas and
where water mass transformation occurs (Sarafanov et al., 2012; García-Ibáñez et al., 2015), the Irminger and
Iceland basins are geographically well placed to monitor temporal changes in the Atlantic MOC (Mercier et al.,
2015), and to determine the rates of $C_{ant}$ penetration to the deep ocean and its consequence for OA.
In this paper, we examine high-quality direct measurements of ocean $CO_2$ chemistry taken from thirteen
cruises conducted across the Irminger and Iceland basins between 1981 and 2015. Previous studies focused on
the $C_{ant}$ uptake and its storage and effect on pH in the Irminger and Iceland basins (e.g., Pérez et al., 2008;
Olafsson et al., 2009; Bates et al., 2012; Vázquez-Rodríguez et al., 2012b). Here we quantify OA for an extended
period and identify its chemical and physical drivers, based on direct measurements.
**2 MATERIALS and METHODS**
**2.1 Datasets**
**2.2.1 Cruise Information**
We used thirteen cruises along the same track across the Irminger and Iceland basins, with the cruise dates
spanning 34 years (1981–2015; Table 1, Fig. 1a). The bottle data were accessed from the merged data product of
the Global Data Analysis Project version 2 (GLODAPv2; Olsen et al., 2016) at
http://cdiac.ornl.gov/oceans/GLODAPv2, except for more recent unpublished data collected during the OVIDE
2012 and 2014 cruises and the 2015 cruise (58GS20150410). The data of the 1991 cruises (64TR91_1 and
06MT18_1) were merged and treated as a single cruise.
**2.2.2 Ocean $CO_2$ chemistry measurements**
At least two variables of the seawater $CO_2$ system were measured on all cruises included in our analyses, but
the measured pairs varied between cruises. The total alkalinity ($A_T$) was analysed by potentiometric titration and





determined by developing either a full titration curve (Millero et al., 1993; Dickson and Goyet, 1994; Ono et al.,
1998) or from single point titration (Pérez and Fraga, 1987; Mintrop et al., 2000), with an overall accuracy of 4
$\mu mol \cdot kg^{-1}$. For samples without direct $A_T$ measurements, it was estimated using a 3D moving window
multilinear regression algorithm (3DwMLR), using potential temperature (θ), salinity, nitrate, phosphate, silicate
and oxygen as predictor parameters (Velo et al., 2013). The total dissolved inorganic carbon (DIC) samples were
analysed with coulometric titration techniques (Johnson et al., 1993), and were calibrated with Certified
Reference Materials (CRMs), achieving an overall accuracy of 2 $\mu mol \cdot kg^{-1}$. The exception to the use of this
analytical technique was the 1981 TTO-NAS (Transient Tracer in the Ocean-North Atlantic Survey) cruise,
where DIC was determined potentiometrically (Bradshaw et al., 1981) and no CRMs were used. The TTO-NAS
DIC measurements were deemed unreliable (Brewer et al., 1986), therefore, the DIC values compiled in the
GLODAPv2 merged data product are those calculated from $pCO_2$ and revised $A_T$ reported by Tanhua and
Wallace (2005). pH was determined either potentiometrically (Dickson, 1993a, b) using pH electrodes or, more
commonly, with a spectrophotometric method (Clayton and Byrne, 1993) using either scanning or diode array
spectrophotometers and m-cresol purple as an indicator. The spectrophotometric pH determination has a typical
precision of 0.0002–0.0004 pH units (Clayton and Byrne, 1993; Liu et al., 2011). However, Carter et al. (2013)
reported an inaccuracy of the spectrophotometric pH determination of 0.0055 pH units. When direct pH
measurements were not performed, it was computed from $A_T$ and DIC using the thermodynamic equations of the
seawater $CO_2$ system (Dickson et al., 2007) and the $CO_2$ dissociation constants of Mehrbach et al. (1973) refitted
by Dickson and Millero (1987). For these calculated pH values, we estimated an uncertainty of 0.006 pH units
by random propagation of the reported $A_T$ and DIC accuracies. The exception to the latter is the 1981 TTO-NAS
cruise, whose DIC problems caused the estimated uncertainty for calculated pH values to be slightly higher
(0.008 pH units). $A_T$ data from the 1981 TTO-NAS cruise were checked against $A_T$ values generated by the
3DwMLR (Velo et al., 2013). $A_T$ values differing by more than two times the standard deviation (confidence
interval; 7 $\mu mol \cdot kg^{-1}$) of the difference between measured $A_T$ and 3DwMLR predicted $A_T$ were replaced with the
predicted $A_T$ value. However, for leg 6 of the 1981 TTO-NAS cruise (which was not analysed by Tanhua and
Wallace (2005)) the limit of substitution for the predicted $A_T$ value was lowered to 4 $\mu mol \cdot kg^{-1}$. Note that the
effect of $A_T$ corrections on pH trends is negligible, since $A_T$ corrections of 4 $\mu mol \cdot kg^{-1}$ lead to pH changes lower
than a thousandth. The pH values reported here are at in situ conditions and on the total scale ($pH_{Tis}$).
**2.2.3 Anthropogenic $CO_2$ (i.e., $C_{ant}$) estimation**
$C_{ant}$ concentrations were estimated using the back-calculation method $\varphi C_T^0$ (Pérez et al., 2008; Vázquez-
Rodríguez, 2009a) that has previously been applied for the entire Atlantic Ocean (Vázquez-Rodríguez et al.,
2009b). Back-calculation methods determine $C_{ant}$ for any sample in the water column as the difference between
DIC concentration at the time of the measurement and the DIC concentration it would have had in preindustrial
times. This is represented as the difference in preformed DIC between the time of observation and the
preindustrial as:
$C_{ant} = DIC_{meas} - \Delta C_{bio} - DIC_{preind} - \Delta C_{diseq}$,                                                          (1)
where the preformed DIC for the time of observation is represented as the measured DIC ($DIC_{meas}$) less any DIC
added to the water due to organic matter remineralisation and calcium carbonate dissolution ($\Delta C_{bio}$), and the
preindustrial preformed concentration is represented by the DIC concentration the water would have if in





equilibrium with the preindustrial atmosphere ($DIC_{preind}$) less any offset from such an equilibrium value, known
as the disequilibrium term ($\Delta C_{diseq}$). The procedure requires DIC and $A_T$ as input parameters, and the empirical
parameterization of the preformed $A_T$ ($A_T^0$) for the computation of the calcium carbonate dissolution and of the
$\Delta C_{diseq}$ term.

120       The $\varphi C_T^0$ method presents two main advantages. First, the spatiotemporal variability of $A_T^0$ is taken into

account. And second, $C_{ant}$ estimation needs no "zero-$C_{ant}$" reference, since the parameterizations of $A_T^0$ and
$\Delta C_{diseq}$ are determined using the subsurface layer as reference for water mass formation conditions (Vázquez-
Rodríguez et al., 2012a). The overall uncertainty of the method has been estimated at 5.2 $\mu mol \cdot kg^{-1}$ (Pérez et al.,
2008; Vázquez-Rodríguez, 2009a).

125       The reproducibilities and uncertainties of the main variables were determined from the deep waters sampled at

Iberian Abyssal Plain during the seven repeats of the OVIDE line, since these waters are expected to be in near-
steady state. The confidence intervals of those samples for each cruise (Table 2) were taken as an estimate of the
uncertainty of the methodologies. The uncertainties of the Apparent Oxygen Utilization (AOU; the difference
between the saturated concentrations of oxygen calculated using the equations of Benson and Krause (1984) and
the measured concentrations of oxygen), $A_T$ and pH on the total scale at 25ºC ($pH_{T25}$) for the seven cruises were
similar. The confidence intervals of $C_{ant}$ (2.4–3.2 $\mu mol \cdot kg^{-1}$) and $pH_{T25}$ (0.004–0.006 pH units) across the seven
cruises are lower than the inherent uncertainty of the $\varphi C_T^0$ estimates (5.2 $\mu mol \cdot kg^{-1}$) and the accuracy of the
spectrophotometric pH measurements (0.0055 pH units), which provides confidence that these data are suitable
for trend determination. The confidence intervals of the $C_{ant}$ estimates are rather similar than in other regions
where $C_{ant}$ has been compared across many cruises (i.e., 2.4 $\mu mol \cdot kg^{-1}$ in the South Atlantic Ocean, Ríos et al.
(2003); 2.7 $\mu mol \cdot kg^{-1}$ in the Equatorial Atlantic Ocean, 24ºN, Guallart et al. (2015); and 2.7 $\mu mol \cdot kg^{-1}$ reported
from a transect along the western boundary of the Atlantic Ocean from 50ºS to 36ºN, Ríos et al. (2015)). The
confidence interval of the mean values of the Iberian Abyssal Plain samples across the seven cruises (last row of
Table 2) was taken as an estimate of the reproducibility of the methodologies. The high reproducibilities, an
order of magnitude lower than the uncertainties, render confidence to the estimated trends.
**2.2 Water mass characterization**

142       Changes in ocean $CO_2$ chemistry were determined for the main water masses in the Irminger and Iceland

basins. These are: (1) Subpolar Mode Water (SPMW); (2) upper and classical Labrador Sea Water (uLSW and
cLSW, respectively); (3) Iceland–Scotland Overflow Water (ISOW) and; (4) Denmark Strait Overflow Water
(DSOW; Fig. 1b). The layers defining the water masses were delimited using potential density following Azetsu-
Scott et al. (2003), Kieke et al. (2007), Pérez et al. (2008) and Yashayaev et al. (2008).

147       To better determine the interfaces between layers and the average value of each variable in each layer, cruise

bottle data were linearly interpolated onto each dbar before determining average variable values, an
improvement with respect to the previous approaches of Pérez et al. (2008, 2010) and Vázquez-Rodríguez et al.
(2012b). Upper layer data (pressure ≤ 100 dbar) were replaced with the mean value in the pressure range 50–100
dbar to reduce the influence of seasonal differences in sampling on the inter-annual trends (Vázquez-Rodríguez
et al., 2012a). Then, the interpolated profiles were divided into the different water mass density intervals (Fig.
1b). Next, the variables were averaged over each density layer on a station by station basis for each cruise.
Finally, the average values in each density layer were determined for each cruise taking into account the





thickness of the layer and the separation between stations. Note that average values of pressure sensitive
parameters, i.e. $pH_{Tis}$, were referred to the mean pressure of the layer over the studied time period to avoid the
effects of the heaving of the water masses due to warming and/or of the sampling strategy over the pH trends.
The average values of the variables for each layer and their confidence intervals can be found in the
Supplementary Table S1.
**2.3 pH deconvolution**
Changes in ocean pH may be brought about by changes in in situ temperature ($T_{is}$), salinity (S), $A_T$, and/or
DIC, of which changes in the latter may be brought about by $C_{ant}$ uptake or by natural processes ($C_{nat}$), such as
remineralisation. $C_{nat}$ is determined as the difference between measured DIC and estimated $C_{ant}$. To estimate how
much each of these altogether five factors contributed to the observed change in pH, we assumed linearity and
decomposed the observed pH changes into these potential drivers according to:
$$\frac{dpH_{Tis}}{dt} = \frac{\partial pH_{Tis}}{\partial T_{is}}\frac{dT_{is}}{dt} + \frac{\partial pH_{Tis}}{\partial S}\frac{dS}{dt} + \frac{\partial pH_{Tis}}{\partial A_T}\frac{dA_T}{dt} + \frac{\partial pH_{Tis}}{\partial DIC}\frac{d(C_{nat}+C_{ant})}{dt},$$ (2)
To estimate $\frac{\partial pH_{Tis}}{\partial var}$ (where $var$ refers to each of the drivers: $T_{is}$, S, $A_T$ and DIC) we calculated the mean $pH_{Tis}$
for each layer and cruise using the real average value of $var$ but keeping the values of the other three drivers
constant and equal to the mean value for the layer over all the cruises. To estimate each $\frac{\partial var}{\partial t}$ term we performed
a linear regression between $var$ and time for each layer.
Trends of all variables involved in Eq. (2) were calculated using the annual interpolation of the observed
values to avoid the bias due to the reduced availability of cruises during the 80's and 90's with respect to the
2000's.
**3 RESULTS AND DISCUSSION**
**3.1 Mean distribution of water mass properties**
The Irminger and Iceland basins in the North Atlantic are characterized by warm and saline surface waters,
and cold and less saline intermediate and deep waters (Fig. 2a,b). The central waters (here represented by the
SPMW layer), which dominates the upper ~700 m, are warmer and saltier in the Iceland basin than in the
Irminger basin, reflecting the water mass transformation that takes place along the path of the North Atlantic
Current (NAC) (Brambilla and Talley, 2008). In particular, the mixing of the SPMW layer with the surrounding
waters while flowing around the Reykjanes Ridge (evident in the salinity distribution; see also García-Ibáñez et
al. (2015)), in conjunction with the air−sea heat loss, results in a colder and fresher SPMW layer in the Irminger
basin. The uLSW and cLSW layers, below the SPMW layer, are warmer and saltier in the Iceland basin due to
their mixing with the surrounding waters during their journey from their formation regions (Bersch et al., 1999;
Pickart et al., 2003; García-Ibáñez et al., 2015). The ISOW layer dominates at depths beneath the cLSW layer.
This layer is warmer and saltier in the Iceland basin, reflecting its circulation. ISOW comes from the Iceland−
Scotland sill and flows southwards into the Iceland basin, where it mixes with the older North Atlantic Deep
Water (NADW). Then, it crosses the Reykjanes Ridge through the Charlie−Gibbs Fracture Zone (Fig. 1a), where
it mixes with the recently ventilated cLSW and DSOW, becoming colder and fresher. In the bottom of the
Irminger basin, a fifth layer is distinguished, DSOW, being the coldest and freshest layer of the section.



The general pattern of $pH_{Tis}$ (Fig. 2c) follows by and large the distribution expected from the surface
production of organic material and remineralisation at depth. The high surface values (> 8.05) are the result of
the withdrawing of DIC by photosynthetic activity, while the values generally decrease with depth down to <
7.95 in the deepest layers, because of the DIC concentration increase resulting from remineralisation. This
overall pattern is disrupted at ~500 m in the Iceland basin by a layer with relatively low $pH_{Tis}$ values (< 7.98),
coinciding with relatively high AOU and DIC values (Fig. 2e,f). This layer could be associated to an area of
slower circulation where the products of the remineralization of the organic matter accumulate. This thermocline
layer could also been influenced by waters of southern origin (Sarafanov et al., 2008), which are advected into
the region by the NAC, whose arrival is closely related with the North Atlantic Oscillation (Desbruyères et al.,
2013). The presence of this low pH layer lowers the average pH of our SPMW layer in the Iceland basin
compared to the Irminger basin (Fig. 3). An opposite pattern is found in the uLSW layer. The water mass
formation occurring in the Irminger basin (Pickart et al., 2003; García-Ibáñez et al., 2015; Fröb et al., 2016;
Piron et al., 2016) transfers recently ventilated low DIC and high pH waters to depth, which causes the mean pH
of uLSW in the Irminger basin to be higher than in the Iceland basin. Finally, the layers that contain the overflow
waters have the lowest pH values. The presence of the older NADW in the ISOW layer in the Iceland basin
decreases the mean pH of this layer here, making it lower than in the Irminger basin.
The surface waters of the section have low DIC values, which rapidly increase when increasing depth (Fig. 2f).
The low DIC values in the uppermost ~200 m are also related to the photosynthetic activity that withdraws DIC
from seawater. Below ~200 m the DIC distribution is almost homogeneous, only disrupted by relatively high
values in the Iceland basin at ~500 m associated with the thermocline layer, and at the bottom, associated with
the old NADW. The gradients in DIC anthropogenic and natural components are much stronger. This is because
the $C_{ant}$ and $C_{nat}$ distributions are anti-correlated. The $C_{ant}$ values are high, close to saturation (80% of saturation),
near the surface and decrease with depth (Fig. 2h), because Cant enters the ocean from the atmosphere. The $C_{nat}$
distribution has an opposite pattern, similar to that of the AOU distribution (Fig. 2e), with low surface values and
high bottom values (Fig. 2g), for reasons discussed above.
The $A_T$ distribution along the section resembles the salinity distribution, with high values associated with the
relatively saline central waters and low and almost homogeneous values in the rest of the section (Fig. 2d). The
exception comes with the deep waters of the Iceland basin, which have among the highest $A_T$ values while
salinity is not extraordinarily high. This reflects the influence of NADW, which contains relatively large
amounts of silicate related to the influence of the Antarctic Bottom Water.

## 3.2 Water mass acidification and drivers

Trends of $pH_{Tis}$ in each layer and basin are presented in Table 3 and Fig. 3. The $pH_{Tis}$ has decreased in all
layers of the Irminger and Iceland basins during the time period of more than 30 years (1981–2015) that is
covered by our data. The trends are stronger in the Irminger basin due to the presence of younger waters. The
rate of OA decreases with depth, except for the DSOW layer that has acidification rates close to those found in
the cLSW layer. This indicates that DSOW is a newly formed water that has recently been in contact with the
atmosphere. Moreover, the acidification rate in the ISOW layer in the Irminger basin is relatively low, which
could be related to the increasing importance on this layer of the relatively old NADW with the diminution in
volume of cLSW since mid-90s (Lazier et al., 2002; Yashayaev, 2007).





The observed rate of $pH_{Tis}$ decrease in the SPMW layer of the Iceland basin (-0.0012 ± 0.0001 pH units·yr$^{-1}$;
Table 3, Fig. 3b) is in agreement with that observed at the Iceland Sea time-series (68ºN, 12.66ºW; Olafsson et
al. (2009, 2010)) for the period 1983–2014 (-0.0014 ± 0.0005 pH units yr$^{-1}$; Bates et al. (2014)). Our rates in the
SPMW layer of both basins are slightly lower than those observed at the Subtropical Atlantic time-series stations
ESTOC (29.04ºN, 15.50ºW; Santana-Casiano et al. (2007), González-Dávila et al. (2010)) for the period 1995–
2014 (-0.0018 ± 0.0002 pH units·yr$^{-1}$; Bates et al. (2014)) and BATS (32ºN, 64ºW; Bates et al. (2014)) for the
period 1983–2014 (-0.0017 ± 0.0001 pH units·yr$^{-1}$; Bates et al. (2014)). However, our rate of $pH_{Tis}$ decrease in
the SPMW layer in the Irminger basin (-0.0013 ± 0.0001 pH units·yr$^{-1}$) is only half of that observed in the sea
surface waters of the Irminger Sea time-series (64.3ºN, 28ºW; Olafsson et al. (2010)) for the period 1983–2014
(-0.0026 ± 0.0006 pH units yr$^{-1}$; Bates et al. (2014)), which is exceptionally high compared to the other time
series summarized here. Comparing with the Pacific Ocean, the OA rates in the Iceland and Irminger basins are
slightly lower than those reported for the Central North Pacific based on data from the time-series station HOT
(22.45ºN, 158ºW; Dore et al. (2009)) for the period 1988–2014 (-0.0016 ± 0.0001 pH units·yr$^{-1}$; Bates et al.
(2014)), but are in agreement with those found by Wakita et al. (2013) in the winter mixed layer at the Subarctic
Western North Pacific (time-series stations K2 and KNOT) for the period 1997–2011 (-0.0010 ± 0.0004 pH
units·yr$^{-1}$).
Vázquez-Rodríguez et al. (2012b) have previously studied the pH changes in the different water masses of the
Irminger and Iceland basins. These authors carried out a pH normalization to avoid potential biases due to
different ventilation stages and rates of each layer, from the different spatial coverage of the evaluated cruises.
The normalized pH values ($pH_N$) for each layer was obtained using multiple linear regressions between the
observed mean $pH_{SWS25}$ (pH at seawater scale and 25ºC) and the observed mean values of θ, salinity, silicate and
AOU, referred to the mean climatological values of θ, salinity, silicate and AOU compiled in WOA05
(http://www.nodc.noaa.gov/OC5/WOA05/pr_woa05.html). This normalization, combined with the lower
temporal coverage (1981–2008) and the fact that they evaluated trends in pH at 25ºC and not at in situ conditions
renders direct comparisons between their and our derived trends difficult.
To infer the causes of the acidification trends reported here, we decomposed the pH trends into their individual
components as described in Sect. 2.2. The results are presented in Table 3. The sum of the pH changes caused by
the individual drivers (in situ temperature, salinity, $A_T$ and DIC) matches the observed pH trends, which renders
confidence to the method.
The temperature changes (Fig. 4a,b) have generally resulted in small to negligible pH declines (Table 3).
Specifically, warming corresponds to a pH decrease of more than 0.0001 pH units·yr$^{-1}$ in the SPMW layer of
both basins and in the LSW layers of the Irminger basin, while the effect of temperature changes on pH in the
other layers is negligible. Temperature driven pH change is larger in the LSW layers in the Irminger than in the
Iceland basin. In the case of the uLSW layer, this is possibly explained by the deep convection occurring in the
Irminger basin (Pickart et al., 2003; García-Ibáñez et al., 2015; Fröb et al., 2016; Piron et al., 2016). In the case
of the cLSW layer, the higher pH changes driven by temperature changes in the Irminger basin could be
explained by the rapid advection of this water mass from the Labrador Sea to this basin (Yashayaev et al., 2007).
The temperature effect on pH evaluated here is mostly thermodynamic. The same applies to the salinity effect,
which however is small to negligible, reflecting that salinity changes in the region (Fig. 4c,d) are insufficiently
large to significantly change pH.



The $A_T$ has increased in all layers (Fig. 5a,b), corresponding to increasing pH (Table 3), which counteracts the
acidification from the $CO_2$ absorption. The contribution from $A_T$ to reduce ocean acidification is significant for
all the layers, except for ISOW of the Irminger basin and uLSW of the Iceland basin (in which $A_T$ trends over
time are not significant; Fig. 5a,b). The $A_T$ increasing trends observed in SPMW may indicate the increasing
presence of waters of subtropical origin (with higher $A_T$) as the subpolar gyre was shrinking since mid-90s (e.g.,
Flatau et al., 2003; Häkkinen and Rhines, 2004; Böning et al., 2006). The $A_T$ effect is evident in the ISOW layer
of the Iceland basin, which can be explained by the circulation and mixing of this layer. As ISOW flows
downstream along the Reykjanes Ridge, it mixes with cLSW and NADW (van Aken and de Boer, 1995;
Fogelqvist et al., 2003). The reduced volume of cLSW since mid-90s (Lazier et al., 2002; Yashayaev, 2007) has
increased the importance of NADW (with high $A_T$; Fig. 2h) in the ISOW layer, making the pH decrease of the
ISOW layer of the Iceland basin lower than in the Irminger basin.
The DIC increase (Fig. 5c,d) is the main cause of the observed pH decreases, and corresponds to pH drops
between -0.00085 and -0.00134 pH units·yr$^{-1}$ (Table 3). The waters in both the Irminger and Iceland basins
gained DIC in response to the increase in atmospheric $CO_2$; the convection processes occurring in these basins
(Pickart et al., 2003; Thierry et al., 2008; de Boisséson et al., 2010; García-Ibáñez et al., 2015; Fröb et al., 2016;
Piron et al., 2016) and in the surrounding ones (i.e., Labrador and Nordic Seas) provide an important pathway for
DIC to pass from the surface mixed layer to the intermediate and deep layers. The effect of the DIC increase on
pH is generally dominated by the anthropogenic component (Table 3). The exception comes with the cLSW
layer of the Irminger basin, where dominates the natural component resulting from the aging of the layer. All
layers have higher $C_{ant}$ increase rates in the Irminger basin than in the Iceland basin (Fig. 6a,b), and therefore
larger pH declines, presumably a result of the proximity of the Irminger basin to the regions of deep water
formation. The highest $C_{ant}$ increase rates are found in the SPMW layer, owing to its direct contact with the
atmosphere, and result in the highest rates of pH decrease. The higher pH drops related to $C_{ant}$ increase found in
the SPMW layer in the Irminger basin compared to those found in the Iceland basin layer, can be related to the
differences in the rise in $C_{ant}$ levels in both basins. In the Irminger basin, the rise in $C_{ant}$ levels of the SPMW
layer correspond to about 85% of the rate expected from a surface ocean maintaining its degree of saturation
with the atmospheric $CO_2$ rise (computed using as reference the measurements of Mauna Loa), while in the
Iceland basin, this rate is about 73% of the expected rate. The lower fraction in the Iceland basin compared to the
Irminger basin is a consequence of the inclusion of the aforementioned poorly ventilated thermocline waters in
our SPMW layer (Fig. 2e,h). Note than none of the $C_{ant}$ trends of the SPMW layers correspond to 100% of the
rate expected from assuming saturation with the atmospheric $CO_2$ rise. This can be explained by the fact that
surface waters $CO_2$ concentration rise lags that of the atmosphere by between two to five years in this region
(Biastoch et al., 2007; Jones et al., 2014). We also note that the temperature and $A_T$ changes impact the pH of
SPMW, decreasing and increasing it, respectively. This could indicate the increasing presence of warmer and
more saline (with higher $A_T$) waters of subtropical origin, which, because $A_T$ effects dominate, in last instance
counteracts the effects of increasing DIC values. Overall this change can be explained as the result of the
contraction of the subpolar gyre that took place since mid-90s (e.g., Flatau et al., 2003; Häkkinen and Rhines,
2004; Böning et al., 2006). Wakita et al. (2013) also found lower than expected acidification rates in the surface
waters of the Pacific Ocean, which they explained as being the consequence of increasing $A_T$. Finally, the strong



influence of anthropogenic component on the pH decrease of the DSOW layer stands out, and is the main agent
of the pH decline in this layer.
The pH change related to $C_{nat}$ changes (Fig. 6c,d) can be interpreted as changes related to ventilation of water
masses and water mass changes (with different $A_T$ and DIC). Higher pH decreases related to $C_{nat}$ changes
indicate lack of ventilation and accumulation of DIC from remineralised organic material. This is clearly the case
for the cLSW layer, where the observed pH decrease is caused by a combination of the effects of $C_{ant}$ and $C_{nat}$.
The greater influence of $C_{nat}$ in the cLSW layer is the result of the aging of this water mass after its last
formation event, in the mid-90s (eg., Lazier et al., 2002; Azetsu-Scott et al., 2003; Kieke et al., 2007; Yashayaev,
2007). $C_{nat}$ also contributes to pH changes in the ISOW layer of the Iceland basin, which is related to the
increasing influence of the relatively old NADW over time due to the decreasing contribution of LSW (Sy et al.,
1997; Yashayaev, 2007; Sarafanov et al., 2010; García-Ibáñez et al., 2015).
**4 CONCLUSIONS**
The progressive acidification of the North Atlantic waters has been assessed from direct observations obtained
over the last three decades (1981–2015), with the greatest pH decreases observed in surface and intermediate
waters. By separating the observed pH change into its main drivers, we corroborate that the observed pH
decreases are a consequence of the oceanic $C_{ant}$ uptake and in addition we find that they have been partially
offset by $A_T$ increases. However, while the $C_{ant}$ concentration of the upper layer roughly keeps up with that
expected from rising atmospheric $CO_2$, the pH decreases at a lower rate than expected from $C_{ant}$ increase. The
increasing arrival of salty and alkaline subtropical waters transported by the NAC to the study region related to
the contraction of the subpolar gyre since mid-90's buffers the acidification caused by the $C_{ant}$ increase in the
upper layer. The acidification rates in intermediate waters are similar to those in the surface waters, and are
caused by a combination of anthropogenic and non-anthropogenic components. The acidification of cLSW due
to the $C_{ant}$ uptake is reinforced by the aging of this water mass from the end of the 1990s onwards. The pH of the
deep waters of the Irminger basin, DSOW, has clearly decreased in response to anthropogenic forcing. We also
observe that water mass warming contributes between 13 and 18% to the pH decrease of the upper and
intermediate waters of the Irminger basin, and 34% to the pH decrease of the upper waters of the Iceland basin.
**Author Contributions**
All authors contributed extensively to the work presented in this paper. M.I.G.-I., A.F.R., H.M., A.O. and
F.F.P. designed the research. M.I.G.-I., P.Z., F.F., L.I.C., A.F.R., H.M., A.O. and F.F.P. analysed the physical
and chemical data. M.I.G.-I. and P.Z. developed the code for processing the data. M.I.G.-I. and F.F.P.
determined the anthropogenic $CO_2$ concentrations, average layer properties and rates, and estimated the
uncertainties. M.I.G.-I. wrote the manuscript and prepared all figures, with contributions from all co-authors.
**Acknowledgements**
We are grateful to the captains, staff and researchers who contributed to the acquisition and processing of
hydrographic data. The research leading to these results was supported through the EU FP7 project





CARBOCHANGE "Changes in carbon uptake and emissions by oceans in a changing climate", which received
funding from the European Commission's Seventh Framework Programme under grant agreement no. 264879.
For this work M.I. Garcia-Ibáñez, A.F. Rios and F.F. Pérez were supported by the Spanish Ministry of Economy
and Competitiveness (BES-2011-045614) through the CATARINA (CTM2010-17141) and BOCATS
(CTM2013-41048-P) projects both co-funded by the Fondo Europeo de Desarrollo Regional 2007-2012
(FEDER). P. Zunino was supported by the GEOVIDE project as well as by IFREMER. L.I. Carracedo was
funded by the University of Vigo, through the Galician I2C Plan for postdoctoral research. H. Mercier was
supported by the French National Centre for Scientific Research (CNRS). F. Fröb and A. Olsen were supported
by a grant from the Norwegian Research Council (SNACS, project 229756/E10).

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



**Table 1: List of hydrographic cruises used in this study (Fig. 1a). P.I. denotes principal investigator, and #St the**
**number of stations selected.**

| Cruise Name | Expocode | Month/Year | Vessel | P.I. | #St | Reference |
|---|---|---|---|---|---|---|
| TTO-NAS L6 | 316N19810821 | 08–09/1981 | *Knorr* | W.J. Jenkins | 11 | Takahashi and Brewer (1986) |
| AR07E | 64TR91_1 | 04–05/1991 | *Tyro* | H.M. van Aken | 12 | Stoll et al. (1996) |
| A01E | 06MT18_1 | 09/1991 | *Meteor* | J. Meincke | 15 | Meincke and Becker (1993) |
| A01E | 06MT30_3 | 11–12/1994 | *Meteor* | J. Meincke | 27 | Koltermann et al. (1996) |
| AR07E | 06MT39_5 | 08–09/1997 | *Meteor* | A. Sy | 32 | Rhein et al. (2002) |
| OVIDE 2002 | 35TH20020610 | 06–07/2002 | *Thalassa* | H. Mercier | 38 | Lherminier et al. (2007) |
| OVIDE 2004 | 35TH20040604 | 06–07/2004 | *Thalassa* | T. Huck | 56 | Lherminier et al. (2010) |
| OVIDE 2006 | 06MM20060523 | 05–06/2006 | *Maria S. Merian* | P. Lherminier | 44 | Gourcuff et al. (2011) |
| OVIDE 2008 | 35TH20080610 | 06–07/2008 | *Thalassa* | H. Mercier | 45 | Mercier et al. (2015) |
| OVIDE 2010 | 35TH20100610 | 06/2010 | *Thalassa* | T. Huck; H. Mercier | 46 | Mercier et al. (2015) |
| CATARINA[a] | 29AH20120623 | 06–07/2012 | *Sarmiento de Gamboa* | A.F. Ríos | 44 | This work |
| GEOVIDE[a] | 35PQ20140517 | 05–06/2014 | *Pourquoi Pas?* | G. Sarthou | 31 | This work |
| 58GS20150410 | 58GS20150410 | 04-05/2015 | *G.O. Sars* | A. Olsen | 10 | Fröb et al. (2016) |

[a]Both CATARINA (http://catarina.iim.csic.es/en) and GEOVIDE (http://www.geovide.obs-vlfr.fr) cruises contain the OVIDE section
(http://wwz.ifremer.fr/lpo/La-recherche/Projets-en-cours/OVIDE), and in the study are referred as OVIDE 2012 and 2014, respectively.



**Table 2: Mean values ± confidence interval of pressure (in dbar), potential temperature (θ, in ºC), salinity, Apparent**
**Oxygen Utilization (AOU, in µmol·kg⁻¹), total alkalinity ($A_T$, in µmol·kg⁻¹), anthropogenic $CO_2$ ($C_{ant}$, in µmol·kg⁻¹) and**
**pH at total scale and 25ºC ($pH_{T25}$) for the bottom waters of the Iberian Abyssal Plain sampled during the seven**
**OVIDE cruises. "n" represents the number of data considered in each cruise. The last row represents the inter-cruise**
**confidence interval (i.e., the confidence interval of the mean values across the seven cruises).**

| Year (n) | Pressure | θ | Salinity | AOU | $A_T$ | $C_{ant}$ | $pH_{T25}$ |
|---|---|---|---|---|---|---|---|
| 2002 (144) | 4205 ± 1052 | 2.182 ± 0.160 | 34.913 ± 0.016 | 86.1 ± 4.0 | 2351 ± 6 | 6.4 ± 2.6 | 7.740 ± 0.006 |
| 2004 (158) | 4263 ± 998 | 2.162 ± 0.150 | 34.908 ± 0.014 | 87.1 ± 2.8 | 2352 ± 6 | 6.2 ± 2.4 | 7.741 ± 0.006 |
| 2006 (132) | 4252 ± 1058 | 2.170 ± 0.164 | 34.913 ± 0.016 | 85.4 ± 3.2 | 2350 ± 6 | 6.2 ± 2.6 | 7.741 ± 0.006 |
| 2008 (125) | 4206 ± 1022 | 2.179 ± 0.150 | 34.911 ± 0.014 | 84.9 ± 3.6 | 2353 ± 8 | 7.0 ± 3.2 | 7.744 ± 0.006 |
| 2010 (131) | 4312 ± 1048 | 2.163 ± 0.154 | 34.908 ± 0.016 | 85.9 ± 3.2 | 2351 ± 6 | 7.0 ± 2.4 | 7.740 ± 0.004 |
| 2012 (102) | 4397 ± 1052 | 2.149 ± 0.154 | 34.909 ± 0.016 | 87.9 ± 3.2 | 2352 ± 6 | 5.1 ± 2.4 | 7.742 ± 0.004 |
| 2014 (54) | 4441 ± 954 | 2.141 ± 0.138 | 34.904 ± 0.014 | 87.4 ± 2.6 | 2353 ± 6 | 5.5 ± 3.0 | 7.743 ± 0.006 |
| | 70 | 0.011 | 0.002 | 0.8 | 0.8 | 0.5 | 0.001 |






**Table 3: Observed temporal changes of pH at total scale and in situ conditions (in situ temperature and pressure; $\frac{dpH_{Tis}}{dt}$ obs) for the main water masses in the Irminger and Iceland basins for the period 1981–2015. pH changes caused by the main drivers (in situ temperature, $T_{is}$; salinity, S; total alkalinity, $A_T$; total dissolved inorganic carbon, DIC; anthropogenic $CO_2$, $C_{ant}$; natural DIC, $C_{nat}$) are also shown, as well as the pH changes result of the sum of the pH changes caused by the individual drivers ($\frac{dpH_{Tis}}{dt}$ model). All the trends are calculated based on the annually interpolated values and are in $10^{-3}$ pH units·yr$^{-1}$. Values in parenthesis are the percentages of the observed pH change explained by each one of its drivers. Confront Fig. 1 for water mass acronyms.**

| | | $\frac{dpH_{Tis}}{dt}$ obs | $\frac{\partial pH_{Tis}}{\partial T_{is}}\frac{dT_{is}}{dt}$ | $\frac{\partial pH_{Tis}}{\partial S}\frac{dS}{dt}$ | $\frac{\partial pH_{Tis}}{\partial A_T}\frac{dA_T}{dt}$ | $\frac{\partial pH_{Tis}}{\partial DIC}\frac{dDIC}{dt}$ | $\frac{\partial pH_{Tis}}{\partial DIC}\frac{dC_{ant}}{dt}$ | $\frac{\partial pH_{Tis}}{\partial DIC}\frac{dC_{nat}}{dt}$ | $\frac{dpH_{Tis}}{dt}$ model |
|---|---|---|---|---|---|---|---|---|---|
| Irminger | SPMW | -1.31 ± 0.08 | -0.24 ± 0.06 (18) | -0.02 ± 0.01 (2) | 0.29 ± 0.05 (-22) | -1.34 ± 0.12 (102) | -1.59 ± 0.10 (121) | 0.24 ± 0.06 (-19) | -1.32 ± 0.14 (100.4) |
| | uLSW | -1.30 ± 0.08 | -0.22 ± 0.02 (17) | $-0.01_7 \pm 0.00_1$ (1) | 0.25 ± 0.01 (-19) | -1.31 ± 0.08 (101) | -1.09 ± 0.13 (84) | -0.22 ± 0.13 (17) | -1.30 ± 0.08 (100.2) |
| | cLSW | -1.06 ± 0.08 | -0.14 ± 0.04 (13) | $-0.01_6 \pm 0.00_4$ (2) | 0.31 ± 0.06 (-29) | -1.22 ± 0.10 (115) | -0.54 ± 0.04 (51) | -0.68 ± 0.11 (64) | -1.07 ± 0.12 (100.4) |
| | ISOW | -0.82 ± 0.08 | 0.03 ± 0.02 (-3) | $0.00_4 \pm 0.00_2$ (0) | -0.01 ± 0.05 (1) | -0.85 ± 0.10 (103) | -0.74 ± 0.06 (89) | -0.11 ± 0.07 (14) | -0.83 ± 0.11 (100.4) |
| | DSOW | -0.91 ± 0.09 | -0.06 ± 0.03 (6) | $-0.00_5 \pm 0.00_2$ (1) | 0.23 ± 0.06 (-25) | -1.09 ± 0.12 (119) | -0.89 ± 0.08 (97) | -0.20 ± 0.07 (22) | -0.92 ± 0.14 (100.7) |
| Iceland | SPMW | -1.18 ± 0.09 | -0.40 ± 0.08 (34) | -0.03 ± 0.01 (2) | 0.44 ± 0.07 (-37) | -1.20 ± 0.11 (102) | -1.25 ± 0.07 (106) | 0.05 ± 0.06 (-4) | -1.19 ± 0.15 (100.6) |
| | uLSW | -0.80 ± 0.05 | 0.03 ± 0.01 (-4) | $0.00_4 \pm 0.00_1$ (-1) | 0.06 ± 0.04 (-7) | -0.90 ± 0.04 (112) | -0.97 ± 0.12 (121) | 0.07 ± 0.09 (-9) | -0.81 ± 0.06 (100.4) |
| | cLSW | -0.76 ± 0.06 | 0.05 ± 0.02 (-6) | $0.00_7 \pm 0.00_2$ (-1) | 0.19 ± 0.05 (-24) | -1.01 ± 0.07 (132) | -0.69 ± 0.05 (91) | -0.31 ± 0.06 (41) | -0.77 ± 0.09 (100.4) |
| | ISOW | -0.61 ± 0.06 | 0.03 ± 0.01 (-4) | $0.00_4 \pm 0.00_1$ (-1) | 0.31 ± 0.07 (-51) | -0.95 ± 0.08 (156) | -0.54 ± 0.07 (89) | -0.41 ± 0.08 (67) | -0.61 ± 0.10 (100.1) |

659





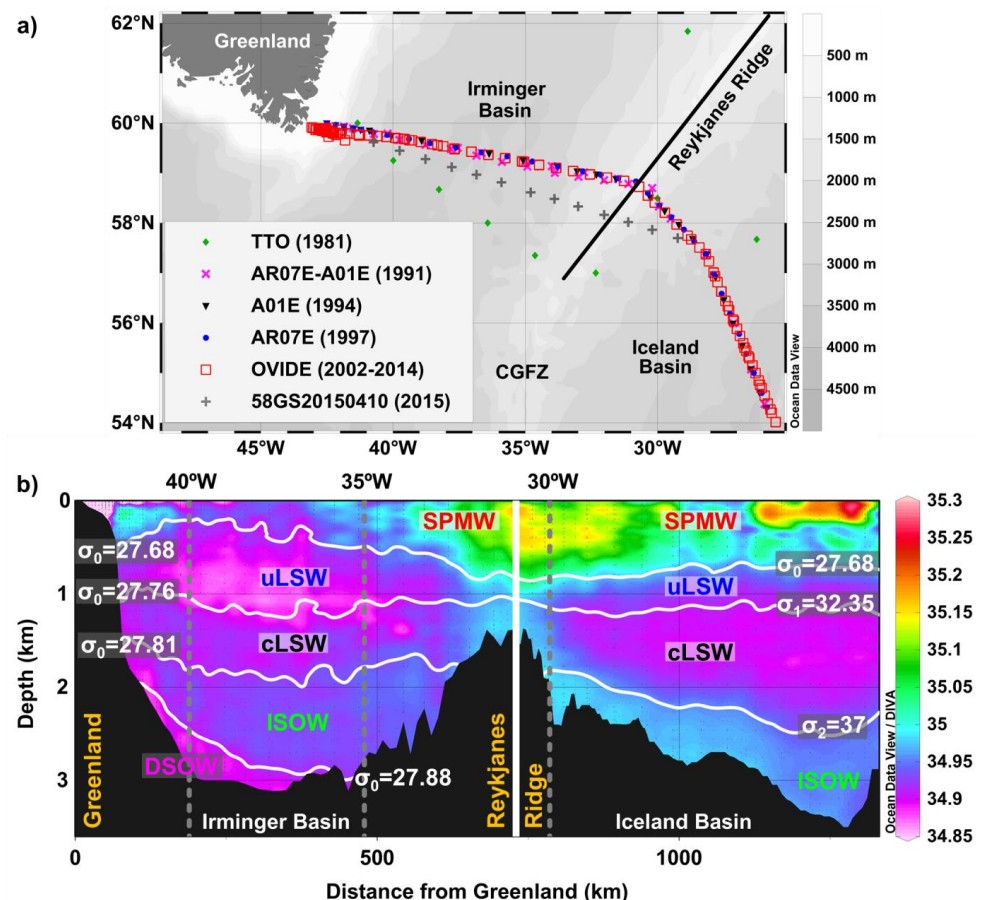

660

Figure 1: (a) Sampling locations of the thirteen cruises used in this study (1981−2015) plotted on bathymetry (500 m intervals). The black line shows the boundary between the Irminger and the Iceland basins constituted by the Reykjanes Ridge. CGFZ = Charlie−Gibbs Fracture Zone. (b) Limits of the layers and basins considered in this study plotted on top of the mean salinity of the sections. The isopycnals delineating the layers are defined by potential density ($\sigma_0$, in kg·m$^{-3}$), and the vertical white line is the limit (Reykjanes Ridge) between the Irminger (left) and Iceland basins (right). The dashed vertical lines represent the Longitude axis marks. The layer acronyms are Subpolar Mode Water (SPMW), upper and classical Labrador Sea Water (uLSW and cLSW, respectively), Iceland−Scotland Overflow Water (ISOW) and Denmark Strait Overflow Water (DSOW).





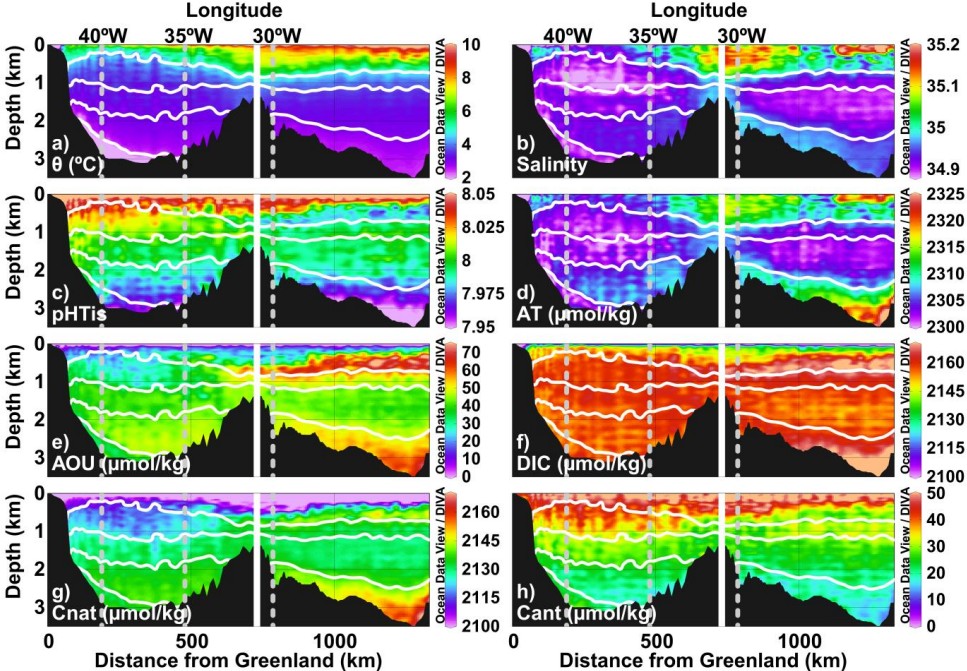

669

Figure 2: Mean distributions along the cruise track, from Greenland (left) to the Iceland basin (right) over study period (1981−2015), for: (a) potential temperature (θ, in ℃), (b) salinity, (c) pH at total scale and in situ conditions (pHTis), (d) total alkalinity (AT, in µmol·kg$^{-1}$), (e) apparent oxygen utilization (AOU, in µmol·kg$^{-1}$), (f) total dissolved inorganic carbon (DIC; in µmol·kg$^{-1}$), (g) natural DIC (Cnat, in µmol·kg$^{-1}$) and (h) anthropogenic $CO_2$ (Cant, in µmol·kg$^{-1}$). The dashed vertical lines represent the Longitude axis marks.





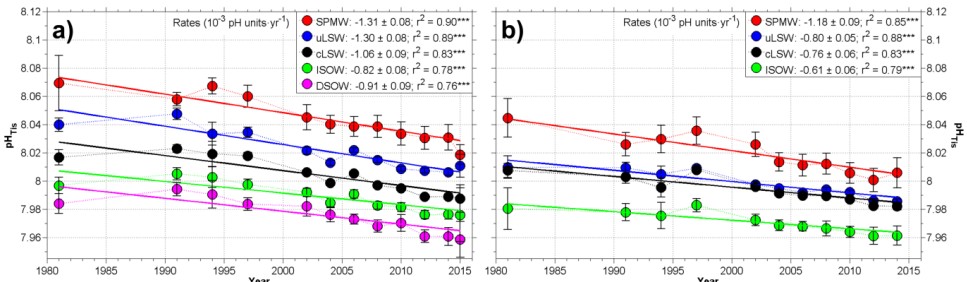

675

**Figure 3: Temporal evolution of average pH at total scale and in situ conditions (pH$_{Tis}$) in the main water masses of the Irminger (a) and Iceland (b) basins, between 1981 and 2015. Each point represents the average pH$_{Tis}$ of a particular layer (SPMW (red dots), uLSW (blue dots), cLSW (black dots), ISOW (green dots) and DSOW (magenta dots)) at the time of each cruise (Table S1). The error bars are two times the error of the mean (2σ = 2x(Standard Deviation)/$\sqrt{N}$, where $N$ is the number of samples of each layer). The inset boxes give the trends (in $10^{-3}$ pH units·yr$^{-1}$) ± standard error of the estimate and the correlation coefficients (r$^2$), resulting from the annually interpolated values. \*\*\* denotes that the trend is statistically significant at the 99% level (p-value < 0.01). Confront Fig. 1 for layer acronyms.**





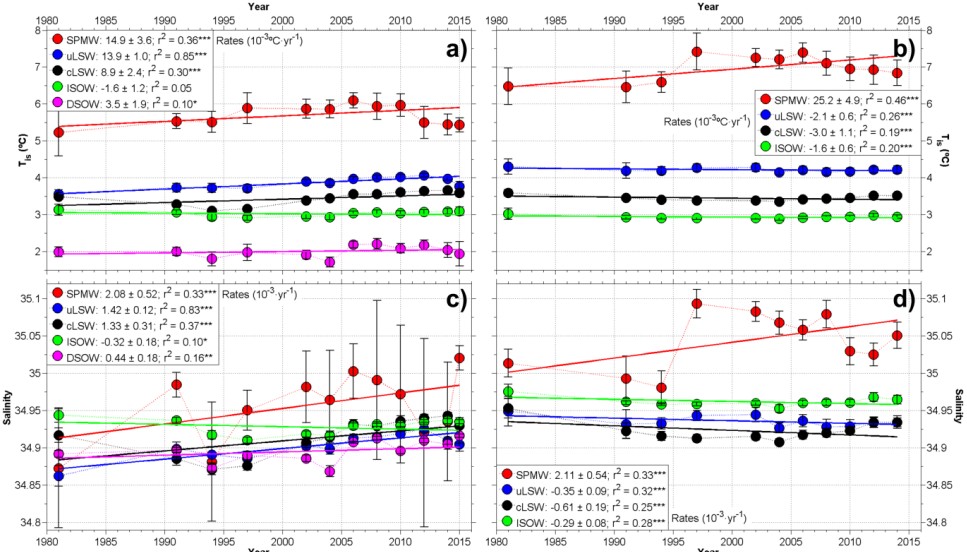

684

**Figure 4: Temporal evolution between 1981 and 2015 of average (a and b) in situ temperature ($T_{is}$, in ºC) and (c and d) salinity in the main water masses of the Irminger (a and c) and Iceland (b and c) basins. Each point represents the average property of a particular layer (SPMW (red dots), uLSW (blue dots), cLSW (black dots), ISOW (green dots) and DSOW (magenta dots)) at the time of each cruise (Table S1). The error bars are 2σ. The inset boxes give the trends (in $10^{-3}$ units·yr$^{-1}$) ± standard error of the estimate and the correlation coefficients ($r^2$), resulting from the annually interpolated values. * denotes that the trend is statistically significant at the 90% level (p-value < 0.1), ** at the 95% level (p-value < 0.05), and *** at 99% level (p-value < 0.01). Confront Fig. 1 for layer acronyms.**





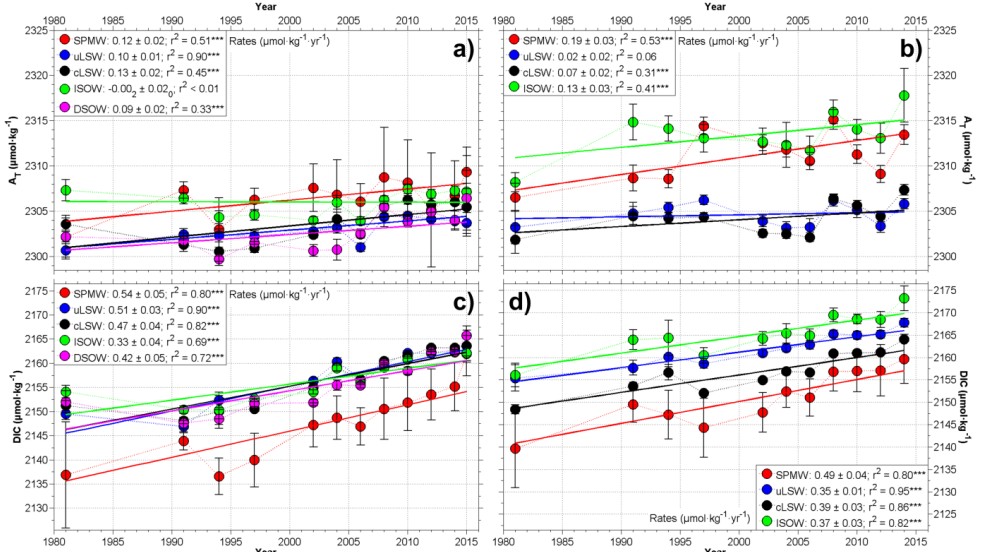

692

**Figure 5: Temporal evolution between 1981 and 2015 of average (a and b) total alkalinity ($A_T$, in µmol·kg$^{-1}$) and (c and d) total dissolved inorganic carbon (DIC, in µmol·kg$^{-1}$) in the main water masses of the Irminger (a and c) and Iceland (b and d) basins. Each point represents the average property of a particular layer (SPMW (red dots), uLSW (blue dots), cLSW (black dots), ISOW (green dots) and DSOW (magenta dots)) at the time of each cruise (Table S1). The error bars are 2σ. The inset boxes give the trends (in µmol·kg$^{-1}$·yr$^{-1}$) ± standard error of the estimate and the correlation coefficients ($r^2$), resulting from the annually interpolated values. \*\*\* denotes that the trend is statistically significant at the 99% level (p-value < 0.01). Confront Fig. 1 for layer acronyms.**





700

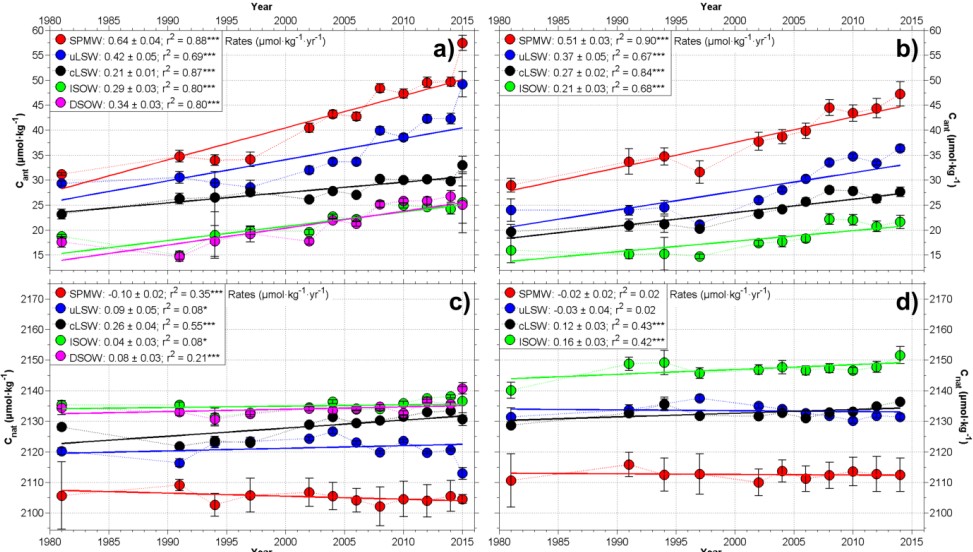

**Figure 6: Temporal evolution between 1981 and 2015 of average (a and b) anthropogenic $CO_2$ ($C_{ant}$, in µmol·kg$^{-1}$) and (c and d) natural DIC ($C_{nat}$ = DIC - $C_{ant}$, in µmol·kg$^{-1}$) values in the main water masses of the Irminger (a and c) and Iceland (b and c) basins. Each point represents the average property of a particular layer (SPMW (red dots), uLSW (blue dots), cLSW (black dots), ISOW (green dots) and DSOW (magenta dots)) at the time of each cruise (Table S1). The error bars are 2σ. The inset boxes give the trends (in µmol·kg$^{-1}$·yr$^{-1}$) ± standard error of the estimate and the correlation coefficients ($r^2$), resulting from the annually interpolated values. \* denotes that the trend is statistically significant at the 90% level (p-value < 0.1), and \*\*\* at the 99% level (p-value < 0.01). Confront Fig. 1 for layer acronyms.**