# Peer review of "Ocean acidification in the Subpolar North Atlantic: rates and mechanisms controlling pH changes"

_Biogeosciences, 2016_

## Referee Comment (RC1) · Anonymous Referee #1 · 13 Apr 2016

General comments

This manuscript deals with the identification and quantification of the main drivers of pH changes in the Iceland and Irminger basins between 1981 and 2015. To do so, high-quality data of 13 research cruises were combined, quality-checked and statistically analysed. Moreover, the contribution of Cant to changes in DIC was calculated and the change in pH was decomposed into five factors that were numerically estimated.

The manuscript is generally well written, and the results are, in my opinion, scientifically sound. My most pressing comments concern the exact methodology on quantifying the main drivers of pH changes, more specifically the use of Equation 2 and the construction of Table 3. I recommend publication after taking into account the following questions and comments.

Specific comments

- title: The title is somewhat too general in my opinion. The manuscript doesn't focus on the whole North Atlantic, just the Irminger and Iceland basins. Also, the controlling factors for the pH change are determined. I'd therefore suggest changing the title into: "Ocean acidification in the Irminger and Iceland basins (of the North Atlantic): mechanisms controlling pH changes' or equivalent.

- l. 63-64: 'Here...measurements' I would change this sentence in various ways. First, 'an extended period' sounds a bit vague. Better state: 'for a 34-year period'. Second, OA is a term used for collective CO2 chemistry changes, while you only quantify the drivers of pH change. This must be made clear here. Third, here would be a good place in the manuscript to already shortly mention how these drivers were identified (i.e. by decomposing the observed pH change into five numerically estimated factors)

- Methods: It is not clear to me if there were cruises where more than two variables were concurrently measured and if so, how these were handled throughout the manuscript in terms of internal consistency. Line 75 implies that such overdetermined stations were present and I'd suggest adding to Table 1 which parameters were measured at each cruise. In the way I understand it, for all samples DIC was measured, and one or both of the variables AT and pH was measured. In the case pH or AT was not measured, it was calculated or estimated from the regression algorithm, respectively. Figures 2c,d,f show these data. The remainder of the calculations (Sects. 2.2 and 2.3), however, only use DIC and AT (i.e. the data presented in Figures 2d and f) and calculate pH from these two variables. If I'm correct, please add this to the manuscript more clearly. If I'm incorrect, please provide a clearer description of which variables were used for which analysis.

- l. 99-100: Is a confidence interval of 2*$\sigma$ or 95% used throughout the manuscript? If so, please add.

- l. 117-119: This statement needs some more explanation. What is 'preformed AT'

and how is it determined?

- l. 131, Table 2: why is pH at 25°C used for this uncertainty analysis, while the remainder of the manuscript deals with values at in situ temperature? Assuming a near-steady state as the authors do, it shouldn't matter which of the two is used.

- l. 150-151: Why is this interval of 50-100 dbar chosen? What is the mixed layer depth in these basins? And is this replacement of the upper layer data also done for the construction of Figure 2? This should be made clear.

- l. 155-157: In combination with the caption of Table S1, this statement is somewhat confusing. Only from this caption I understood that pH in Table S1 (and also Figure 3, and dpH/dt_obs in Table 3) was calculated from DIC and TA rather than interpolated from measured pH values. This is important information that needs to be part of the main text. Moreover, I'm curious as to whether the authors have tried correcting the measured pH values for the mean pressure of the layer cf. Millero (1995) and how this compared to the average pH estimated using this method.

- l. 161: Does a change in salinity also include the effect due to a change in borate? If so, what salinity – borate relationship is used? This information should also be added to Sect. 2.1.1.

- l. 166, eq (2): Why is $\delta$pH/$\delta$DIC not split into $\delta$pH/$\delta$Cant and $\delta$pH/$\delta$Cnat? This is one of the few points of the manuscript that is really unclear to me. The authors should be able to vary Cant while keeping Cnat constant and thus calculate these factors separately.

- l. 167-173: It is important that the authors clearly state how they calculated the data presented in Table 3. Therefore this section needs some improvement. I assume that dvar/dt is calculated based on the regression lines presented in Figures 4-6 (which are based on annually interpolated data). It remains unclear, however, how $\delta$pH/$\delta$var is estimated. It is important to realise that $\delta$pH/$\delta$var is not a constant parameter, its value calculated from the 1981 data will be substantially different from that calculated

based on the 2015 data (see, e.g. Riebesell et al., 2009). What is the 'mean pH' the authors refer to in l. 167? (and, similarly, what is the 'real average value of var'?) Is it the mean pH of a certain layer of the 34-year period or the mean pH of that layer for each (annually interpolated) year? I assume it is the latter, and therefore it would be very interesting to see the temporal evolution of all the partial differentials over time. Could the authors add these data to the manuscript or supplementary information? Presenting the temporal evolution of these 'buffer factors' can also aid the discussion in Sect 3.2.

- l. 212: An explanation is required of what the 'saturation of Cant' involves. I saw later that it is explained in l. 294-297, so I would move this explanation forward to Sect 3.1. In terms of Eq. 1, would a saturated Cant mean that $\Delta$Cbio and $\Delta$Cdiseq are 0?

- l. 236-240: I believe that the authors should elaborate on why their pH decrease in the Irminger basin is so different from the values presented by Bates et al. (2014), rather than just stating that the Bates et al. (2014) value 'is exceptionally high compared to the other time series summarized here'. The work of Bates et al. (2014) is also done on seasonally detrended time series and the obtained rate of change is statistically significant (P<0.01), so the fact that the results of both analyses are so different should be the basis for an interesting scientific discussion. Bates et al. (2014) link the high rate of pH decrease in the Irminger Sea directly to the high rate of pCO2 increase at this site; it would be interesting to read the authors' opinion on this.

- l. 240-245: I don't feel that the comparison with the Pacific adds much to the manuscript.

- l. 252-254: Perhaps the authors could additionally evaluate their trends at 25°C for comparison with this study, as it would be very interesting to see the differences resulting from the various data interpolation methods.

- l. 267: Mostly or fully thermodynamic? What other, non-thermodynamic effect could be there?

- l. 296: Why are data from Mauna Loa used here and not from a more closely located measurement station?

- l. 304: Perhaps clarify that even though salinity also changes (in concurrence with AT), the salinity effect on pH is still negligible.

- l. 311-312: What about changes in the production / respiration balance? Could they also be responsible for the observed Cnat changes?

- l. 333-334: It is not physically meaningful to talk about percentages when discussing contributions to a change in pH, as pH is on a logarithmic scale. Use absolute values or percentages of changes in [H+] instead. This also applies to Table 3.

- Figure 2: How is this figure constructed, what is the order of interpolation here? Were the data linearly interpolated over time before the mean was calculated at each sampling point? Or was the mean calculated using the spatio-temporally integrated data? This information needs to be added to the figure caption and/or the Method section.

- Table 3: How are the confidence intervals calculated here? Also, be more explicit about the difference between dpH/dt_obs and dpH/dt_model throughout the manuscript (see also comment on Eq. (2)).

Technical corrections

- l. 37: shouldn't 1750 be 1850?

- l. 43-44: I feel that the number of references is too high here, since biological effects are not studied in this manuscript

- l. 53-54: also here the number of relevant references could be reduced, though it is less problematic here than in the previous section

- l. 62: remove 'the' in 'the Cant uptake'

- l. 67: should be '2.1.1' (same applies to '2.1.2' on l. 74 and '2.1.3' on l. 105)

- l. 76: remove 'the' in 'the total alkalinity'

- l. 113: replace 'less' by 'minus' (also in l. 116)

- l. 166, Eq (2): add the subscript 'model' to the left hand side, to be consistent with the right column of Table 3 (distinguishing more clearly between dpH/dt_obs and dpH/dt_model could be done throughout the manuscript)

- l. 169: replace '$\delta var/\delta dt$' with 'dvar/dt', these are ordinary differentials.

- l. 288: move 'dominates' to the end of the sentence.

- l. 304: 'in last instance' is not very clear. Do you mean 'in a net sense'?

- l. 325: remove 'however', this sentence is not contradictory with the previous one

- Table 1: for each cruise, add which carbonate system parameters are measured

- Table 3: why are the last digits in the column describing the salinity effect on pH presented with subscripts?

- General comment on the figures: be consistent with the amount of significant digits on the colour bar and/or y-axis (e.g. 35.3 vs. 35.25 for Figure 1b). This applies to all figures in the manuscript

- Figure 1a: the colour scheme is not very clear, the light-dark gradient could be more extreme

- Figures 3-6: some general comments on these figures: please use different symbols for the different water masses, this makes the figures readable on black & white. Also, add the title of the basin on top of the figure (Irminger basin left column, Iceland basin right column), this makes the figures more accessible without having to read the caption. Finally, the dotted lines (annually interpolated values) are hardly visible.

- Figures 4 and 6: '(b and c)' should be replaced with '(b and d)'

**References**

Riebesell U., Körtzinger A. and Oschlies A. (2009) Sensitivities of marine carbon fluxes to ocean change. Proc. Natl. Acad. Sci. U. S. A. 106, 20602–20609.

---

## Referee Comment (RC2) · Anonymous Referee #2 · 29 Apr 2016

Review of "Ocean acidification in the North Atlantic: controlling mechanisms" by Maribel I. García-Ibáñez, Patricia Zunino, Friederike Fröb, Lidia I. Carracedo, Aida F. Ríos, Herlé Mercier, Are Olsen, Fiz F. Pérez, as submitted to Biogeosciences Discussions in Feb. 2016.

This manuscript analyses a rich set of data of CO2-system measurements made in the subpolar North Atlantic Ocean over a period of 34 years, aiming to update the existing record of strong acidification of these waters and, newly, to apportion the total acidification to a suite of driving processes. The paper is excellently readable, modest and concise. I have no conceptual reservations with its general methodology. Minor criticism I voice is the rather strong likeliness to earlier work performed by some of the authors, which however is overridden by the new focus on attribution. Overall, I feel the manuscript warrants publication, with only minor revisions. Desired changes

mostly relate to lack of explicitness in the methods sections, and the confusing (or even confused?) use of statistical concepts, which is where this review focuses.

General comments:

– Consider adding "Subpolar" to North Atlantic in the title.

– I believe your results are occasionally strongly affected by the TTO data (particularly in the Irminger basin), conceivably worsened by your time-interpolation performed to 'provide weight to old cruises'. I recommend publication of your results with exclusion of TTO data, or least mention in the text of how such exclusion would affects results.

– The sections plots of Fig1 and Fig2 are unrealistically noisy. The captions suggests the "mean distributions" are plotted, but these are not averages, but rather all data of all cruises thrown into a single section, with inappropriately short influence radii for the contouring (or whatever the equivalent terminology is for DIVA gridding). They thus represent not natural spatial heterogeneity, but temporal aliasing. This leads to disturbingly jittery artifacts (particularly evident in Fig2d as blue/purple/pink patchwork). Consider either contouring true averages, or simply increasing the influence radii (i.e., smooth it more).

– Consider adding a visually catchy and informative summarizing section plot (one for pHobs, or perhaps one per pH-driver), showing per water mass the rate of pH change. In each, surface layers would most red, as would DSOW, with intermediate layers slightly lower, and Iceland on average lower than Irminger.

Specific comments:

– Consider capturing some more cruise details in your Table 1. For instance, please tabulate the type of measurements performed on each cruise (which had pH directly, which calculated it – i.e., you lines ∼75–100). What is the consequence of the rather seriously sounding, but nonchalantly made remark in line 91 "However, Carter et alk reported a pH inaccuracy of 0.0055"? Is that a positive or negative bias? Systematic

for everyone or just for them? Do you compensate?

– line 113: less => minus

– line 120: "advantages" relative to what? $\Delta C^*$?

– line 121: I can't follow. The suggestion is that no Cant-free reference waters are required, but it's not clear why that is. Consider explaining more clearly or not at all and only keeping the reference to VR2012).

– line 131: explain why you consider 0.0055 the "accuracy" of the pH measurements. Again, if Carter thinks this is a /systematic/ error of the method, that would not affect detectability of trends.

– I find the use of statistical terminology confusing. The terms "standard deviation", "confidence interval" and seem to be used loosely or even interchangeably while they each have a clearly defined use case. (Line 99-100 seems to suggest that you equate "two standard deviations" to "confidence interval"). If the use of these terms is nonetheless is correct then certainly the employed confidence level should be mentioned to make sense of the stated confidence intervals. I particularly object to referring to the standard deviation of depths in a defined depth (or density) range as the 'confidence interval' of depths (first column of T1).

– I hold the whole of line 125-140 to constitute a slight misuse of statistical numbers. The reasoning here seems to be "the spread between the means of cruises is smaller than the spread within each cruise, and thus we believe we can detect trends between cruises". Although the closeness in cruise means is certainly comforting, that alone does not make for detectability of trends. It would at best provide a lower bound for the detectability of trends (i.e., trends within the ranges given in T1 would go unnoticed but might nonetheless exist). Consider adding a small statement that indicates these results are suggestive of high quality, and try to avoid suggesting to provide evidence thereof.

– line 150: this 'replacement' process is a little rash. I can imagine ignoring these shallow data, but simply overwriting them with data that has less sensitivity to seasonality without providing a compelling case for doing so is not warranted. I do not believe that ignoring the 100m surface layer would yield a vastly different result to what you now got. If that is indeed so, I recommend using that ignore-approach, to avoid the suggesting that you're fudging.

– line 154: please be specific in how you "take into account thickness and separation". I presume that the average of tall profiles with large spacing to east and west get higher weight in average-of-averages? Do you have a specific reason for not using the alternative approach of averaging per layer the 'grid boxes' of Fig2? Presumably you did not grid the data for analysis but only for figure making.

– I'm not clear on what you've done here. I agree that pressure-adjustment here is necessary, although heaving of even 500 m would not even produce a pH shift larger than 0.02 units. However, you'd do this also simply to reduce the range of pH values within each (non-horizontal) water mass. I would, however, believe the proper procedure to be (i) "recalc pH for each sample in water mass to pH at the single mean depth of that watermass (for the cruise or the whole dataset, that shouldn't matter much) and then (ii) calculate average of these recalculated pH values. This might be what you did, but the way I read it, you first calculated the average pH, and then shifted that average pH to the 'correct' depth. If so, consider redoing more appropriately, or explain for daft readers like myself why the used approach *is* appropriate.

– line 157: remove last four words "over the pH trends"

– I can't fully comprehend what the approach is that was followed in section 2.3. The idea is clear "keep all but one parameter constant and see how pH changes. The sensitivity of pH to an increase in DIC would be sharper in 2015 than in 1981. Is that accounted for in the method? Specify the calculation routine you used.

– Consider restructuring 2.3 into a distinct paragraph for the determination of time

trends and one for inferring strength of individual drivers. Your TABLE3 mentions the "sum of drivers" or "model", which terminology is nowhere used in the text, please harmonize. Also, T3 separates influence of Cnat and Cant, but Eq2 does not.

– line 168: "real average value" => "observed linear trend" (???)

– TABLE1: I believe "confidence interval" here is "standard deviation", or is it truly CI? Then state the confidence level. I'm not sure the CI of the average of averages, or however one would call the last row, has any statistical meaning – why not simply provide SD in that row? You use pH25 in table 1, while stating in the text that pHisT and pH25 are not easily compared – why the sudden use of pH25 here?

– FIGURE1: some of the contour intervals have at sig1 or sig2 label, while caption and text suggest cutoffs were based on sig0.

– I generally very much like your other (time-tested) figures. Perhaps increase coverage of fig 1a to provide a view of distance to land on eastern extent of section.

– Consider moving 3.1 to the introduction.

– line 209: "almost homogenous" – sections plots suggest otherwise, see earlier comment on influence radii

– line 211: "because they are correlated" – that relationship is not causal, please rephrase.

– line 212: can you qualify that "80%" in light of the mentioned $\Delta$Cdiseq? Is this what one would expect?

– line 239: can you speculate on the possible causes for the supposedly spuriously high rate of pH decrease observed by Bates et al at IrmSTS?

– line 244: you mention a tropical Pacific time series station, and contrast it with your work and a subpolar Pacific TSS, latter two match nicely. Add a brief sentence attributing that contrast.

– line 246-254: your statement "renders direct comparison difficult" does not stand up to scrutiny. Recalculating pH to different temperatures does not changes the slope of a pH trend. That is, slopes can be compared (i.e., VR12's Fig3ab vs your Fig3ab), even if absolute values cannot. I recommend more work is made of this comparison, particularly if results between studies differ.

– line 322: if anything, these are 4 decades (80s 90s 00s and 10s). Consider "34-year period" or similar

– line 323 (and likely elsewhere): "separate and increase into its drivers" is slightly sloppy English. Consider rephrasing

– line 325: "However" => "thus". Reduced rate of decrease of pH is what one expects with increasing alk.

– line 327: "salty" => "saline"

– line 333: consider "observe" => "infer". There's too many interpretative steps involved to call this "observe"

---

## Author Comment (AC2) · 23 May 2016

*We thank referee #2 for the helpful comments. We have addressed the referee's concerns as explained below.*

General comments:

– Consider adding "Subpolar" to North Atlantic in the title.

*Added.*

– I believe your results are occasionally strongly affected by the TTO data (particularly in the Irminger basin), conceivably worsened by your time-interpolation performed to 'provide weight to old cruises'. I recommend publication of your results with exclusion of TTO data, or least mention in the text of how such exclusion would affects results.

*Thank you for your suggestion. We recalculated all the trends excluding TTO data and the difference between pH trends with and without TTO were significant. Hence, we decided to exclude the TTO data from our study, with all the changes that it entails.*

– The sections plots of Fig1 and Fig2 are unrealistically noisy. The captions suggests the "mean distributions" are plotted, but these are not averages, but rather all data of all cruises thrown into a single section, with inappropriately short influence radii for the contouring (or whatever the equivalent terminology is for DIVA gridding). They thus represent not natural spatial heterogeneity, but temporal aliasing. This leads to disturbingly jittery artifacts (particularly evident in Fig2d as blue/purple/pink patchwork). Consider either contouring true averages, or simply increasing the influence radii (i.e., smooth it more).

*Thank you for your suggestions. You are right, what we did is plotting the data of all cruises in a single section, rather than averaging the data. Since the purpose of describing Figure 2 is giving a general view of the properties along this section, we think that describing a single cruise is enough. For this reason, we decided to describe the general distribution of the main variables along the section using the 2004 cruise as reference. This cruise represents the mean year of the studied period. In this way we avoid interpolating the data from all cruises to a single grid and then averaging them to build Figure 2, with all the errors that this would entail.*

– Consider adding a visually catchy and informative summarizing section plot (one for pHobs, or perhaps one per pH-driver), showing per water mass the rate of pH change. In each, surface layers would most red, as would DSOW, with intermediate layers slightly lower, and Iceland on average lower than Irminger.

*Thank you for your suggestion. We added the suggested pHobs figure to the supplementary information (Figure S1).*

Specific comments:

– Consider capturing some more cruise details in your Table 1. For instance, please tabulate the type of measurements performed on each cruise (which had pH directly, which calculated it – i.e., you lines ~75–100). What is the consequence of the rather seriously sounding, but nonchalantly made remark in line 91 "However, Carter et al reported a pH inaccuracy of 0.0055"? Is that a positive or negative bias? Systematic for everyone or just for them? Do you compensate?

*We added an extra column that specifies the measurements performed on each cruise. What we wanted to highlight with the sentence in line 91 is the fact that although it is possible to achieve high reproducibility in pH measurements, all the measurements will have an inherent uncertainty of 0.0055 due to the uncertainty in the determination of the constants of the tris-buffer. Therefore, the 0.0055 quantity is an uncertainty that affects all pH measurements, which we cannot compensate. We have clarified this point by changing the highlighted sentence to: "However, Carter et al. (2013) reported an inherent uncertainty of spectrophotometric pH determinations of 0.0055 pH units, associated to the tris-buffer used for calibration". (See also answer to comment about line 131).*

– line 113: less => minus

*Corrected*

– line 120: "advantages" relative to what? ΔC*?

*Yes. We added the following statement at the end of the sentence: "relative to the previous method proposed by Gruber et al. (1996)".*

– line 121: I can't follow. The suggestion is that no Cant-free reference waters are required, but it's not clear why that is. Consider explaining more clearly or not at all and only keeping the reference to VR2012).

*We changed the statement to make it clearer: "And second, the parameterizations of $A_T^0$ and $\Delta C_{diseq}$ are determined using the subsurface layer as reference (Vázquez-Rodríguez et al., 2012a), where the age of the water parcel and, therefore, its $C_{ant}$ concentration is estimated using CFC measurements (Waugh et al., 2006)".*

– line 131: explain why you consider 0.0055 the "accuracy" of the pH measurements. Again, if Carter thinks this is a /systematic/ error of the method, that would not affect detectability of trends.

*You are right; this uncertainty will affect all pH measurements and, therefore, will not affect the detectability of trends. What we wanted to highlight with this statement is that the data we are using have high reproducibility (higher than the accuracy of the measurements) and thus are suitable for determining trends. We added the following statement to the manuscript: "The high reproducibility, an order of magnitude better than the uncertainty (0.0055 pH units, Carter et al. (2013)), is suggestive of high quality data". (See also answer to comment about line 91).*

– I find the use of statistical terminology confusing. The terms "standard deviation", "confidence interval" and seem to be used loosely or even interchangeably while they each have a clearly defined use case. (Line 99-100 seems to suggest that you equate "two standard deviations" to "confidence interval"). If the use of these terms is nonetheless is correct then certainly the employed confidence level should be mentioned to make sense of the stated confidence intervals. I particularly object to referring to the standard deviation of depths in a defined depth (or density) range as the 'confidence interval' of depths (first column of T1).

*Thank you for your comment; we noticed that we were using the term 'confidence interval' incorrectly. In lines 99-100 the term 'confidence interval' was not correctly used and, therefore, it was deleted. In the revised manuscript we*

*changed the term 'confidence interval' with 'standard deviation'. We, therefore, change the numbers presented in the tables accordingly. We also eliminated the standard deviation of the pressure data in Table 2 as you suggested. We only used the confidence intervals in figures 3-6, where we defined what we consider as confidence interval (2x(standard deviation)/$\sqrt{N}$, where N is the number of samples), which corresponds to a 95% confidence interval since our samples are independent.*

– I hold the whole of line 125-140 to constitute a slight misuse of statistical numbers. The reasoning here seems to be "the spread between the means of cruises is smaller than the spread within each cruise, and thus we believe we can detect trends between cruises". Although the closeness in cruise means is certainly comforting, that alone does not make for detectability of trends. It would at best provide a lower bound for the detectability of trends (i.e., trends within the ranges given in T1 would go unnoticed but might nonetheless exist). Consider adding a small statement that indicates these results are suggestive of high quality, and try to avoid suggesting to provide evidence thereof.

*Thank you for your suggestions. We agree with your comment. We changed lines 139-140 as follows: "The high reproducibility, an order of magnitude better than the uncertainty (0.0055 pH units, Carter et al. (2013)), is suggestive of high quality data. Using these standard deviations for the seven cruises, and taking into account the 25 years considered in this study, the threshold of detectability of pH trends at 95% of confidence is 0.00012 pH units·yr$^{-1}$, which renders confidence to the estimated trends".*

– line 150: this 'replacement' process is a little rash. I can imagine ignoring these shallow data, but simply overwriting them with data that has less sensitivity to seasonality without providing a compelling case for doing so is not warranted. I do not believe that ignoring the 100m surface layer would yield a vastly different result to what you now got. If that is indeed so, I recommend using that ignore-approach, to avoid the suggesting that you're fudging.

*Thank you for your suggestion. We followed it by removing the photic layer (pressure < 75 dbar) from our study, so as to avoid the seasonality effect. This depth was determined by the depth of the seasonal nutrients drawdown. Thus, we replaced sentence in line 150 by the following sentence: "To reduce the influence of seasonal differences in sampling on the inter-annual trends, only samples with pressure ≥ 75 dbar were considered. The 75 dbar level was determined by the depth of the seasonal nutrients drawdown along the section".*

– line 154: please be specific in how you "take into account thickness and separation". I presume that the average of tall profiles with large spacing to east and west get higher weight in average-of-averages? Do you have a specific reason for not using the alternative approach of averaging per layer the 'grid boxes' of Fig2? Presumably you did not grid the data for analysis but only for figure making.

*You are right when highlighting the different weighting given to the profiles in function of their spacing, but, intuitively, horizontally gridding with a linear interpolation would give the same result as the method used in our study.*

*We did not grid the data horizontally but vertically, and this vertical gridding was taken into account for both figure building and calculations. Averages were then performed in an area-weighted basis, so that accounting for the irregular spacing between stations. This was performed by using a trapezoidal integration, which*

*produces the same results as the horizontally gridding, which consumes more computational time.*

– I'm not clear on what you've done here. I agree that pressure-adjustment here is necessary, although heaving of even 500 m would not even produce a pH shift larger than 0.02 units. However, you'd do this also simply to reduce the range of pH values within each (non-horizontal) water mass. I would, however, believe the proper procedure to be (i) "recalc pH for each sample in water mass to pH at the single mean depth of that watermass (for the cruise or the whole dataset, that shouldn't matter much) and then (ii) calculate average of these recalculated pH values. This might be what you did, but the way I read it, you first calculated the average pH, and then shifted that average pH to the 'correct' depth. If so, consider redoing more appropriately, or explain for daft readers like myself why the used approach *is* appropriate.

> *Sorry for the misunderstanding of our methodology. What we did was calculate an average pH for each layer and year from the average values of DIC and $A_T$, and referred to the mean pressure of the layer over the studied time period. We rewrote this explanation in the manuscript: "The exception comes with $pH_{Tis}$, which is pressure sensitive, and for which we needed to define a unique reference pressure to remove pressure effects due to varying sampling strategies. $pH_{Tis}$ was calculated using the layer average values of DIC and $A_T$ for the considered year but using the time-averaged pressure of the layer over the studied time period as reference pressure". We have also corroborated that the methodology you propose and the methodology that we applied give the same results. This is because the variability of the physicochemical properties within each layer is relatively low, and therefore the system is linear. Following this line of reasoning we added the following sentence in the first paragraph of section 2.2: "The advantage of working in layers is the relatively low variability of the physical and chemical properties within the layers, allowing us to assume linearity in the ocean $CO_2$ system". We, then, decided to keep our methodology since it is simpler and working with averaged DIC and $A_T$ (conservative) is desirable instead of working with recalculated pH (not conservative) values for each sample and then averaging them. Besides, the recalculated pH for each sample that you propose will not reproduce the pH shown in Figure 2c.*

– line 157: remove last four words "over the pH trends"

> *Done.*

– I can't fully comprehend what the approach is that was followed in section 2.3. The idea is clear "keep all but one parameter constant and see how pH changes. The sensitivity of pH to an increase in DIC would be sharper in 2015 than in 1981. Is that accounted for in the method? Specify the calculation routine you used.

> *We are aware that the 'buffer factors' change with time and that the system is not linear. However, since we split the studied region in different layers, the range of variation of the parameters within each layer during the studied time period is small (the ± of Table S1 gives insights about this small variability), and then, we can assume linearity. Also because of the small range of variation of the parameters within each layer, the range of variation of the 'buffer factors' is also small, and we can neglect their change when calculating the change in pH due to each of the proposed controlling mechanisms. I am enclosing an example about the change of 'δpH/δDIC' in the SPMW layer of the Irminger basin. The red*

*points show the 'δpH/δDIC' values for each year, and the black point is the 'δpH/δDIC' used in the study. As you can see, the change of 'δpH/δDIC' over time is negligible, and the value used in the study is the mean value for the period 1991-2015.*

[Figure]

*We added the following statement to clarify this question in the manuscript: "Given that the variability of the physicochemical properties within each layer is relatively low (see standard deviations of the averaged values in Table S1), we can assume that these derivatives are constant over the studied time period and use a constant derivative value for each layer".*

– Consider restructuring 2.3 into a distinct paragraph for the determination of time trends and one for inferring strength of individual drivers. Your TABLE3 mentions the "sum of drivers" or "model", which terminology is nowhere used in the text, please harmonize. Also, T3 separates influence of Cnat and Cant, but Eq2 does not.

*We included a table in the Supplementary Information that shows the strength of individual drivers (Table S2). Since time trends are represented in figures 4-6, rebuilding section 2.3 would involve citing figures 4-6 before figure 2, which we think this is not a logical order for figures in the manuscript. However, we added the following statement to the results section to clarify where the reader can find the derivative values: "The values of each term of $\frac{\partial pH_{Tis}}{\partial var}$ and $\frac{dvar}{dt}$ (where $var$ refers to each of the drivers) described in Sect. 2.2 can be found in the Supplementary Table S2 and in Figs. 4-6, respectively".*

*We changed dpH/dt_model by dpH/dt_total, and introduced this new term to Eq. (2) and Table 3. We also added some explanatory comments throughout the manuscript to distinguish more clearly between dpH/dt_model and dpH/dt_total.*

*We slightly changed Eq. (2) to clearly indicate the separation between $C_{ant}$ and $C_{nat}$:* $\left(\frac{dpH_{Tis}}{dt}\right)_{total} = \frac{\partial pH_{Tis}}{\partial T_{is}}\frac{dT_{is}}{dt} + \frac{\partial pH_{Tis}}{\partial S}\frac{dS}{dt} + \frac{\partial pH_{Tis}}{\partial A_T}\frac{dA_T}{dt} + \frac{\partial pH_{Tis}}{\partial DIC}\left(\frac{dC_{ant}}{dt} + \frac{dC_{nat}}{dt}\right)$

– line 168: "real average value" => "observed linear trend" (???)

> *What we meant is the average value calculated for each layer and cruise. We changed the sentence to the following to make this clearer: "To estimate $\frac{\partial pH_{Tis}}{\partial var}$ (where $var$ refers to each of the drivers: $T_{is}$, $S$, $A_T$ and $DIC$) we calculated a $pH_{Tis}$ for each layer and year using the layer average value of $var$ for each year but keeping the values of the other drivers constant and equal to the time-average value for the layer over the studied time period".*

– TABLE1: I believe "confidence interval" here is "standard deviation", or is it truly CI? Then state the confidence level. I'm not sure the CI of the average of averages, or however one would call the last row, has any statistical meaning – why not simply provide SD in that row? You use pH25 in table 1, while stating in the text that pHisT and pH25 are not easily compared – why the sudden use of pH25 here?

> *We believe you are referring to Table 2. We changed 'confidence interval' by 'standard deviation' and we changed the results accordingly. The purpose of this table is to give an estimate of the reproducibility of the analysis and calculation methods, which we take as an indicative of the goodness of the data for trend analysis. Regarding the scale of pH, since trends are performed using pH at in situ conditions, we followed your suggestion and we changed the pH reported in Table 2 by pH at in situ conditions.*

– FIGURE1: some of the contour intervals have at sig1 or sig2 label, while caption and text suggest cutoffs were based on sig0.

> *Sorry for the confusion, sigθ would refer to all the sigma levels, either referenced to 0, 1000 or 2000 dbar. To make this clearer we changed the figure caption by "(referenced to 0 dbar, $\sigma_0$; 1000 dbar, $\sigma_1$; and 2000 dbar, $\sigma_2$; all in kg·m$^{-3}$)"*

– I generally very much like your other (time-tested) figures. Perhaps increase coverage of fig 1a to provide a view of distance to land on eastern extent of section.

> *We updated Figure 1a following your suggestions. We increased the eastward extension of the map.*

– Consider moving 3.1 to the introduction.

> *We appreciate your suggestion. However, and despite the qualitative nature of this section, we think it fits better in the results section rather than in the introduction, since it is a description based on our data. Besides, it provides useful information for understanding the sections coming next. In view of this, we decided to keep this section as it is.*

– line 209: "almost homogenous" – sections plots suggest otherwise, see earlier comment on influence radii

> *After changing Figure 2 as suggested in the previous comment regarding this figure, now the statement is true.*

– line 211: "because they are correlated" – that relationship is not causal, please rephrase.

> *You are right. We eliminated this statement, so that, in our opinion, the explanation of property distributions is improved.*

– line 212: can you qualify that "80%" in light of the mentioned ΔCdiseq? Is this what one would expect?

> *The existence of a saturation lower than 100% in the surface layer is expectable for an oceanic uptaken of $C_{ant}$ (see Matsumoto and Gruber, 2006). Therefore, a disequilibrium in the $C_{ant}$ content with respect to the atmospheric $C_{ant}$ is expected.*
>
> *Matsumoto, K. and Gruber, N.: How accurate is the estimation of anthropogenic carbon in the ocean? An evaluation of the ΔC\* method, Global Biogeochem. Cycles, 19, doi:10.1029/2004GB002397, 2005.*

– line 239: can you speculate on the possible causes for the supposedly spuriously high rate of pH decrease observed by Bates et al at IrmSTS?

> *We added the explanation given by Bates et al. (2014): "Bates et al. (2014) linked the high acidification rate found at the Irminger Sea time-series to the high rate of increase in DIC ($1.62 \pm 0.35$ µmol·kg$^{-1}$·yr$^{-1}$) observed at this site, which is almost twice our rate of increase in DIC ($0.64 \pm 0.07$ µmol·kg$^{-1}$·yr$^{-1}$, Fig. 5c). This is based on data from only one site, further north than our section, and indicates that spatial variations are substantial in this region". We do not have enough information to elucidate other possible explanations.*

– line 244: you mention a tropical Pacific time series station, and contrast it with your work and a subpolar Pacific TSS, latter two match nicely. Add a brief sentence attributing that contrast.

> *Now that we are not including the TTO cruise data in our study, our trends are in agreement with the trends found in the tropical Pacific and are slightly higher than the trends found in the subpolar Pacific. We changed the manuscript accordingly to these changes, and we added the following explanation to the low trends found in the subpolar Pacific: "Wakita et al. (2013) attributed the lower than expected pH trends to an increasing $A_T$ trend".*

– line 246-254: your statement "renders direct comparison difficult" does not stand up to scrutiny. Recalculating pH to different temperatures does not changes the slope of a pH trend. That is, slopes can be compared (i.e., VR12's Fig3ab vs your Fig3ab), even if absolute values cannot. I recommend more work is made of this comparison, particularly if results between studies differ.

> *The fact that precludes comparing trends is not the difference in the temperature at which pH is referenced, but the fact that Vazquez-Rodriguez et al. (2012b) normalized pH values to potential temperature, salinity, silicate and AOU in WOA05. This normalization eliminates part of the influence of these parameters (potential temperature, salinity, silicate and AOU) on the pH trends. This is why direct comparison between their pH trends and our pH trends is difficult. However, we added some text comparing the trends reported by both works: "This normalization, combined with the different temporal coverage (1981–2008), causes the rates reported by Vazquez-Rodriguez et al. (2012b) differ from those obtained in the present work. The $pH_N$ trends reported for the SPMW and uLSW layers of the Irminger basin and for the ISOW layer of the Iceland basin are very similar to our $pH_{Tis}$ trends for these layers. However, the $pH_N$ trends reported by Vazquez-Rodriguez et al. (2012b) for the cLSW layer in both basins and for the ISOW layer in the Irminger basin are significantly different from our $pH_{Tis}$ trends for these layers, but are very similar to pH changes derived from $C_{ant}$ changes*

($\frac{\partial pH_{Tis}}{\partial DIC}\frac{dC_{ant}}{dt}$ *in Table 3). In the case of the DSOW layer, the* $pH_N$ *trend is also in agreement with* $\frac{\partial pH_{Tis}}{\partial DIC}\frac{dC_{ant}}{dt}$ *trends. This suggests that the normalization carried out by Vazquez-Rodriguez et al. (2012b) could remove some of the impact of the natural component (represented here by $C_{nat}$) over pH changes, essentially due to the use of AOU in the normalization".*

– line 322: if anything, these are 4 decades (80s 90s 00s and 10s). Consider "34-year period" or similar

*We changed it by "25-year period", since now we are not considering the TTO cruise.*

– line 323 (and likely elsewhere): "separate and increase into its drivers" is slightly sloppy English. Consider rephrasing

*We changed the sentence into: "From the study of the main drivers of the observed pH changes…".*

– line 325: "However" => "thus". Reduced rate of decrease of pH is what one expects with increasing alk.

*Changed.*

– line 327: "salty" => "saline"

*Changed.*

– line 333: consider "observe" => "infer". There's too many interpretative steps involved to call this "observe"

*Changed.*

---

## Author Response (AR1)

**Reply to Referee #1**

We thank referee #1 for the helpful comments. We have addressed the referee's concerns as explained below.

- title: The title is somewhat too general in my opinion. The manuscript doesn't focus on the whole North Atlantic, just the Irminger and Iceland basins. Also, the controlling factors for the pH change are determined. I'd therefore suggest changing the title into: "Ocean acidification in the Irminger and Iceland basins (of the North Atlantic): mechanisms controlling pH changes' or equivalent.

Following your suggestion and suggestions from Referee #2 we changed the title into "Ocean acidification in the Subpolar North Atlantic: rates and mechanisms controlling pH changes".

- l. 63-64: 'Here. . .measurements' I would change this sentence in various ways. First, 'an extended period' sounds a bit vague. Better state: 'for a 34-year period'. Second, OA is a term used for collective CO2 chemistry changes, while you only quantify the drivers of pH change. This must be made clear here. Third, here would be a good place in the manuscript to already shortly mention how these drivers were identified (i.e. by decomposing the observed pH change into five numerically estimated factors)

Following your suggestions we changed this sentence into "Here we quantify the pH change for a 25-year period and identify its chemical and physical drivers by decomposing the observed pH change into five numerically estimated factors (temperature, salinity, alkalinity, anthropogenic  $CO_2$  and non-anthropogenic  $CO_2$ ), all based on direct measurements". Note that the timeframe of the study has decreased to 25 years since, following suggestion of Referee #2, TTO data was excluded from our study.

- Methods: It is not clear to me if there were cruises where more than two variables were concurrently measured and if so, how these were handled throughout the manuscript in terms of internal consistency. Line 75 implies that such overdetermined stations were present and I'd suggest adding to Table 1 which parameters were measured at each cruise. In the way I understand it, for all samples DIC was measured, and one or both of the variables AT and pH was measured. In the case pH or AT was not measured, it was calculated or estimated from the regression algorithm, respectively. Figures 2c,d,f show these data. The remainder of the calculations (Sects. 2.2 and 2.3), however, only use DIC and AT (i.e. the data presented in Figures 2d and f) and calculate pH from these two variables. If I'm correct, please add this to the manuscript more clearly. If I'm incorrect, please provide a clearer description of which variables were used for which analysis.

Sorry for the confusion. We added the suggested column to Table 1. We also added some explanatory comments in the 2.1.2 section (Ocean  $CO_2$  chemistry measurements). The first sentence was changed to: "The twelve cruises selected for our study have high-quality measurements of the seawater  $CO_2$  system variables (Table 1)", since there are cruises with DIC measurements only. We also added the following sentence to clarify how we obtain DIC values when not available: "For the cruises where direct DIC measurements had not been performed, it was computed from  $A_T$  and pH using the thermodynamic equations of the seawater  $CO_2$  system (Dickson et al., 2007) and the  $CO_2$  dissociation constants of Mehrbach et al. (1973) refitted by Dickson and Millero (1987)." - 1. 99-100: Is a confidence interval of  $2*\sigma$  or 95% used throughout the manuscript? If so, please add.

Thank you for your comment. We noticed that we were using incorrectly the term 'confidence interval'. In lines 99-100 the term 'confidence interval' was not correctly used and, therefore, it was deleted. In the revised manuscript we have replaced the term 'confidence interval' with 'standard deviation'. We only used the confidence intervals in figures 3-6, where we defined what we use as confidence interval  $(2x(standard deviation)/\sqrt{N}, where N is the number of samples), which is a 95% confidence interval since the samples are independent.$

- l. 117-119: This statement needs some more explanation. What is 'preformed AT' and how is it determined?

We added the following explanation at the end of the paragraph: "The  $A_T^0$  is based on the concept of potential alkalinity ( $PA_T = A_T + NO_3 + PO_4$ ) and is defined as  $A_T^0 = PA_T - (NO_3^0 + PO_4^0)$  (Vázquez-Rodríguez et al., 2012a), where  $NO_3^0$  and  $PO_4^0$  are the preformed nitrate and phosphate concentrations, respectively.  $NO_3^0$  and  $PO_4^0$  are determined as  $NO_3^0 = NO_3 - AOU/R_{ON}$  and  $PO_4^0$ =  $PO_4 - AOU/R_{OP}$ . In the former equations AOU stands for Apparent Oxygen Utilisation, which is the difference between the saturated concentrations of oxygen calculated using the equations of Benson and Krause (1984) and the measured concentrations of oxygen;  $R_{ON}$  and  $R_{OP}$  are the Redfield ratios proposed by Broecker (1974)".

- l. 131, Table 2: why is pH at 25°C used for this uncertainty analysis, while the remainder of the manuscript deals with values at in situ temperature? Assuming a near-steady state as the authors do, it shouldn't matter which of the two is used.

The purpose of Table 2 is to give an estimate of the reproducibility of the analysis and calculation methods. This reproducibility gives insights about the goodness of the data for trend analysis. Since trends are determined using pH at in situ conditions, we followed your suggestion and we changed the pH reported in Table 2 to pH at in situ conditions.

- l. 150-151: Why is this interval of 50-100 dbar chosen? What is the mixed layer depth in these basins? And is this replacement of the upper layer data also done for the construction of Figure 2? This should be made clear.

Following the comment of Referee #2 about the same issue, we have slightly changed the methodology and now we removed the data from the photic layer (pressure < 75 dbar). We also removed this upper data to construct figures 1 and 2.

- 1. 155-157: In combination with the caption of Table S1, this statement is somewhat confusing. Only from this caption I understood that pH in Table S1 (and also Figure 3, and dpH/dt\_obs in Table 3) was calculated from DIC and TA rather than interpolated from measured pH values. This is important information that needs to be part of the main text. Moreover, I'm curious as to whether the authors have tried correcting the measured pH values for the mean pressure of the layer cf. Millero (1995) and how this compared to the average pH estimated using this method.

We are sorry for the confusion. In fact, pH in Table 3 and Figure 3, and in the newly added Table S2 of the Supplementary Information, was calculated as you state. Therefore we changed the statement of lines 155-157 by the following

statement, hoping that it is clearer now: "The exception comes with  $pH_{Tis}$ , which is pressure sensitive, and for which we needed to define a unique reference pressure to remove pressure effects due to varying sampling strategies.  $pH_{Tis}$  was calculated using the layer average values of DIC and  $A_T$  for the considered year but using the time-averaged pressure of the layer over the studied time period as reference pressure".

Regarding the pressure correction proposed by Millero (1995), we did not perform this correction to our data. Millero's corrections were developed to correct the data before computer power was sufficient enough to perform the proper calculations. In our study, we took the advantage of the CO2SYS software (Lewis and Wallace, 1998; van Heuven et al., 2011), making not necessary the use of the corrections proposed by Millero (1995).

Lewis, E. and Wallace, D. W. R.: Program developed for CO2 system calculations, 30 ORNL/CDIAC-105, Carbon Dioxide Information Analysis Center, Oak Ridge National Laboratory, Oak Ridge, TN, 1998. van Heuven, S., Pierrot, D., Rae, J. W. B., Lewis, E., and Wallace, D. W. R.: MATLAB program

developed for CO2 system calculations, ORNL/CDIAC-105b, Carbon Dioxide Information Analysis Center, Oak Ridge National Laboratory, TN, 15 doi:10.3334/CDIAC/otg.CO2SYS\_MATLAB\_v1.1, 2011.

- l. 161: Does a change in salinity also include the effect due to a change in borate? If so, what salinity – borate relationship is used? This information should also be added to Sect. 2.1.1.

Yes, the change in salinity includes the effect due to borate change since when changing salinity  $A_T$  and DIC are maintained constant. Regarding the salinityborate relationship, we used the constants recommended by Glodap v2 and by the 'Guide to Best Practices for Ocean CO2 Measurements' of Dickson et al. (2007), which are the constants of Uppström (1974). We added the following clarifying statements in reference to this doubt: "Changes in temperature and salinity influence the equilibrium constants of the oceanic CO2 system. Additionally, changes in salinity influence the borate concentration, whose influence is taken into account by the relationship proposed by Uppström (1974)".

Uppström, L.R.: Boron/chlorinity ratio of deep-sea water from the Pacific Ocean, Deep-Sea Res., 21, 161–162, doi:10.1016/0011-7471(74)90074-6, 1974.

- l. 166, eq (2): Why is  $\delta pH/\delta DIC$  not split into  $\delta pH/\delta Cant$  and  $\delta pH/\delta Cnat$ ? This is one of the few points of the manuscript that is really unclear to me. The authors should be able to vary Cant while keeping Cnat constant and thus calculate these factors separately.

We did not split  $\delta pH/\delta DIC$  because the change in pH per unit of DIC is going to be the same if the DIC molecule is  $C_{ant}$  or is  $C_{nat}$ . Hence, we think it is not necessary to split the 'buffer factor' between  $C_{ant}$  and  $C_{nat}$ . We added the following clarifying statement: "Note that sensitivity of  $pH_{Tis}$  to changes in  $C_{ant}$  is the same as the sensitivity to changes in  $C_{nat}$  since both are DIC, and, therefore, only  $\frac{\partial pH_{Tis}}{\partial DIC}$  is necessary".

- 1. 167-173: It is important that the authors clearly state how they calculated the data presented in Table 3. Therefore this section needs some improvement. I assume that dvar/dt is calculated based on the regression lines presented in Figures 4-6 (which are based on annually interpolated data). It remains unclear, however, how  $\delta pH/\delta var$  is estimated. It is important to realise that  $\delta pH/\delta var$  is not a constant parameter, its value calculated from the 1981 data will be substantially different from that calculated based on the 2015 data (see, e.g. Riebesell et al., 2009). What is the 'mean pH' the authors refer to in 1. 167? (and, similarly, what is the

'real average value of var'?) Is it the mean pH of a certain layer of the 34-year period or the mean pH of that layer for each (annually interpolated) year? I assume it is the latter, and therefore it would be very interesting to see the temporal evolution of all the partial differentials over time. Could the authors add these data to the manuscript or supplementary information? Presenting the temporal evolution of these 'buffer factors' can also aid the discussion in Sect 3.2.

As stated on lines 171-173, trends of all variables involved in Eq. 2 (and therefore in Table 3) were calculated using the annually interpolated data. As you stated, dvar/dt were calculated based on regression lines presented in Figures 4-6. Regarding the terms ' $\delta pH/\delta var$ ', we calculated the pH for each layer and year (also for the interpolated years, without cruises) keeping all but the parameter in question constant and equal to the mean value for the layer over the study time period. With an example, if we want to calculate  $\delta pH/\delta S$ , what we do is to calculate a pH for each layer and year using the average S value for each layer and year but the mean values for each layer over the studied time period (1991-2015) for the rest of variables (T,  $A_T$  and DIC). To make this clearer in the text, we changed the sentence in lines 167-173 by: "To estimate  $\frac{\partial pH_{Tis}}{\partial var}$  (where var refers to each of the drivers:  $T_{is}$ , S,  $A_T$  and DIC) we calculated a pHTis for each layer and year using the layer average value of var for each year but keeping the values of the other drivers constant and equal to the time-average value for the layer over the studied time period".

We are aware that the 'buffer factors' change with time and that the system is not linear. However, since we split the studied region in different layers, the range of variation of the parameters within each layer during the studied time period is small (the  $\pm$  of Table S1 gives insights about this small variability), and then, we can assume linearity. Also because of the small range of variation of the parameters within each layer, the range of variation of the 'buffer factors' is also small, and we can neglect their change when calculating the change in pH due to each of the proposed controlling mechanisms. I am enclosing an example about the change of ' $\delta pH/\delta DIC$ ' in the SPMW layer of the Irminger basin. The red points show the ' $\delta pH/\delta DIC$ ' values for each year, and the black point is the ' $\delta pH/\delta DIC$ ' used in the study. As you can see, the change of ' $\delta pH/\delta DIC$ ' over time is negligible, and the value used in the study is the mean value for the period 1991-2015.

We added the following statement to clarify this question in the manuscript: "Given that the variability of the physicochemical properties within each layer is relatively low (see standard deviations of the averaged values in Table S1), we can assume that these derivatives are constant over the studied time period and use a constant derivative value for each layer".

- l. 212: An explanation is required of what the 'saturation of Cant' involves. I saw later that it is explained in l. 294-297, so I would move this explanation forward to Sect 3.1. In terms of Eq. 1, would a saturated Cant mean that  $\Delta$ Cbio and  $\Delta$ Cdiseq are 0?

We added a similar explanation to that found in lines 294-297 ("approximately 80% of the  $C_{ant}$  concentration expected from a surface ocean in equilibrium with the atmospheric  $CO_2$ "). Regarding the doubts about Eq. 1, the term  $\Delta C_{diseq}$  is never 0 since it depends on the conditions of the water mass at the time of its formation. It is known that water masses are not in complete equilibrium with the atmospheric  $CO_2$  concentration when formed (see Matsumoto and Gruber, 2005). Besides, a disequilibrium in the  $C_{ant}$  content with respect to the atmospheric  $C_{ant}$  is expected, since ocean in uptaken  $C_{ant}$ . Therefore, the surface layer is never saturated in  $C_{ant}$ . To sum up, a water mass can be saturated in oxygen, and hence  $\Delta C_{bio}$  is 0, but  $\Delta C_{diseq}$  is never 0.

Matsumoto, K. and Gruber, N.: How accurate is the estimation of anthropogenic carbon in the ocean? An evaluation of the  $\Delta C^*$  method, Global Biogeochem. Cycles, 19, doi:10.1029/2004GB002397, 2005.

- 1. 236-240: I believe that the authors should elaborate on why their pH decrease in the Irminger basin is so different from the values presented by Bates et al. (2014), rather than just stating that the Bates et al. (2014) value 'is exceptionally high compared to the other time series summarized here'. The work of Bates et al. (2014) is also done on seasonally detrended time series and the obtained rate of change is statistically significant (P<0.01), so the fact that the results of both analyses are so different should be the basis for an interesting scientific discussion. Bates et al. (2014) link the high rate of pH decrease in the Irminger Sea directly to the high rate of pCO2 increase at this site; it would be interesting to read the authors' opinion on this.

We added the explanation given by Bates et al. (2014): "Bates et al. (2014) linked the high acidification rate found at the Irminger Sea time-series to the high rate of increase in DIC ( $1.62 \pm 0.35 \mu mol \cdot kg^{-1} \cdot yr^{-1}$ ) observed at this site, which is almost twice our rate of increase in DIC ( $0.64 \pm 0.07 \mu mol \cdot kg^{-1} \cdot yr^{-1}$ , Fig. 5c). This is based on data from only one site, further north than our section, and indicates that spatial variations are substantial in this region". We do not have enough information to elucidate other possible explanations.

- 1. 240-245: I don't feel that the comparison with the Pacific adds much to the manuscript.

Following suggestions of Referee #2, we are going to keep this comparison, adding some extra information.

- 1. 252-254: Perhaps the authors could additionally evaluate their trends at 25°C for comparison with this study, as it would be very interesting to see the differences resulting from the various data interpolation methods.

The fact that precludes comparing trends is not the difference in the temperature to which pH is referenced, but the fact that Vazquez-Rodriguez et al. (2012b)

normalized pH values to climatological potential temperature, salinity, silicate and AOU (WOA values). This normalization eliminates part of the influence of these parameters (potential temperature, salinity, silicate and AOU) on the pH trends. This is why direct comparison between their pH trends and our pH trends is difficult. However, we added some text comparing the trends reported by both works: "This normalization, combined with the different temporal coverage (1981–2008), causes the rates reported by Vazquez-Rodriguez et al. (2012b) differ from those obtained in the present work. The  $pH_N$  trends reported for the SPMW and uLSW layers of the Irminger basin and for the ISOW layer of the Iceland basin are very similar to our  $pH_{Tis}$  trends for these layers. However, the  $pH_N$ trends reported by Vazquez-Rodriguez et al. (2012b) for the cLSW layer in both basins and for the ISOW layer in the Irminger basin are significantly different from our  $pH_{Tis}$  trends for these layers, but are very similar to pH changes derived from  $C_{ant}$  changes  $\left(\frac{\partial pH_{Tis}}{\partial DIC}\frac{dC_{ant}}{dt}\right)$  in Table 3). In the case of the DSOW layer, the  $pH_N$  trend is also in agreement with  $\frac{\partial pH_{Tis}}{\partial DIC}\frac{dC_{ant}}{dt}$  trends. This suggests that the normalization carried out by Vazquez-Rodriguez et al. (2012b) could remove some of the impact of the natural component (represented here by  $C_{nat}$ ) over pH changes, essentially due to the use of AOU in the normalization".

- 1. 267: Mostly or fully thermodynamic? What other, non-thermodynamic effect could be there?

We eliminated mostly, because referee is right, all the effect is thermodynamic.

- l. 296: Why are data from Mauna Loa used here and not from a more closely located measurement station?

Stations closer to the location of our study area, e.g., Mace Head, do not have the same time-coverage as our study. Therefore, we changed the  $pCO_2$  data to the 'Globally averaged marine surface annual mean data' from the NOAA (ftp://aftp.cmdl.noaa.gov/products/trends/co2/co2 annmean gl.txt), since  $pCO_2$  in the atmosphere is almost homogeneous worldwide, and hence the degree of saturation does not change.

- l. 304: Perhaps clarify that even though salinity also changes (in concurrence with AT), the salinity effect on pH is still negligible.

We added the following clause to line 304: "(as stated before, the effect of salinity change on pH is negligible)".

- l. 311-312: What about changes in the production / respiration balance? Could they also be responsible for the observed Cnat changes?

Now that we are not taking into account the photic layer (see answer to comment about 'l. 150-151'), changes in  $C_{nat}$  cannot be brought about by changes in production. Besides, what we refer to as 'changes related to the ventilation of water masses' involves both changes in the renewal of the water mass with upper waters and changes in the respiration. To make this clear, we added the following statement when describing the general pattern of  $C_{nat}$  distribution (section 3.1): "The  $C_{nat}$  distribution has an opposite pattern, with low surface values and high bottom values (Fig. 2g), similar to that of the AOU distribution (Fig. 2e), since  $C_{nat}$  is linked to the ventilation of water masses, i.e., respiration and renewal of the water mass". - 1. 333-334: It is not physically meaningful to talk about percentages when discussing contributions to a change in pH, as pH is on a logarithmic scale. Use absolute values or percentages of changes in [H+] instead. This also applies to Table 3.

We are aware that pH is on a logarithmic scale, but for the small range of pH change to which we are working, we can consider that pH follows a linear scale. That is why the use of percentages is meaningful. Besides, we found that the percentage of change in terms of  $[H^+]$  is exactly the same than that calculated in terms of pH (for the reasons explained above). Finally, the scientific community is working in terms of pH. For all these reasons we decided to keep this work in terms of pH. We added the following statement to the 2.3 section: "Due to the small range of pH change to which we are working and to the relatively low pH variability within each layer, we can consider that pH follows a linear scale instead of a logarithmic scale. This causes that the contributions of each of the terms considered in Eq. (2) to pH change are equivalent to the contributions in terms of  $[H^+]$ ".

- Figure 2: How is this figure constructed, what is the order of interpolation here? Were the data linearly interpolated over time before the mean was calculated at each sampling point? Or was the mean calculated using the spatio-temporally integrated data? This information needs to be added to the figure caption and/or the Method section.

Figure 2 was built by loading all the cruise data to a single section plot on the ODV, and then a DIVA gridding was performed. Since the purpose of describing Figure 2 is giving a general view of the properties along this section, we think that describing a single cruise is enough. For this reason, we decided to describe the general distribution of the main variables along the section using the 2004 cruise as reference. This cruise represents the mean year of the studied period. In this way we avoid interpolating the data from all cruises to a single grid and then averaging them to build Figure 2, with all the errors that this would entail.

- Table 3: How are the confidence intervals calculated here? Also, be more explicit about the difference between dpH/dt\_obs and dpH/dt\_model throughout the manuscript (see also comment on Eq. (2)).

The  $\pm$  were calculated by propagation of errors of the slopes of each of the derivatives. We also added some explanatory comments throughout the manuscript to distinguish more clearly between dpH/dt\_obs and dpH/dt\_model, changing the later by dpH/dt\_total.

Technical corrections

- 1. 37: shouldn't 1750 be 1850?

We do not think so. We have chosen 1750 according to Caldeira and Wickett (2005).

Caldeira, K. and Wickett, M.E.: Ocean model predictions of chemistry changes from carbon dioxide emissions to the atmosphere and ocean, J. Geophys. Res., 110, C09S04, doi:10.1029/2004JC002671, 2005.

- l. 43-44: I feel that the number of references is too high here, since biological effects are not studied in this manuscript

We eliminated three references.

- l. 53-54: also here the number of relevant references could be reduced, though it is less problematic here than in the previous section

We eliminated two references.

- 1. 62: remove 'the' in 'the Cant uptake'

Done.

- 1. 67: should be '2.1.1' (same applies to '2.1.2' on 1. 74 and '2.1.3' on 1. 105)

Thank you for catching this typo. We made the corresponding corrections.

- 1. 76: remove 'the' in 'the total alkalinity'

Done.

- l. 113: replace 'less' by 'minus' (also in l. 116)

Done.

- l. 166, Eq (2): add the subscript 'model' to the left hand side, to be consistent with the right column of Table 3 (distinguishing more clearly between dpH/dt\_obs and dpH/dt\_model could be done throughout the manuscript)

We changed dpH/dt\_model by dpH/dt\_total, and introduced this new term to Eq. (2) and Table 3. We also added some explanatory comments throughout the manuscript to distinguish more clearly between dpH/dt\_obs and dpH/dt\_total.

- l. 169: replace 'δvar/δt' with 'dvar/dt', these are ordinary differentials.

Thank you for catching this typo. We made the correction.

- 1. 288: move 'dominates' to the end of the sentence.

Done.

- 1. 304: 'in last instance' is not very clear. Do you mean 'in a net sense'?

Yes, we meant 'in a net sense'. We changed the expression.

- 1. 325: remove 'however', this sentence is not contradictory with the previous one

We changed 'however' by 'thus' as suggested by Referee #2.

- Table 1: for each cruise, add which carbonate system parameters are measured

Added

- Table 3: why are the last digits in the column describing the salinity effect on pH presented with subscripts?

They were presented because they are not significant. However, we decided to present them without subscript.

- General comment on the figures: be consistent with the amount of significant digits on the colour bar and/or y-axis (e.g. 35.3 vs. 35.25 for Figure 1b). This applies to all figures in the manuscript.

For figures built with ODV (i.e., Figures 1 and 2) is not possible to change the amount of significant digits in the color bars. For the remaining figures, the amount of significant digits has been updated.

- Figure 1a: the colour scheme is not very clear, the light-dark gradient could be more extreme

Figure updated following your suggestions.

- Figures 3-6: some general comments on these figures: please use different symbols for the different water masses, this makes the figures readable on black & white. Also, add the title of the basin on top of the figure (Irminger basin left column, Iceland basin right column), this makes the figures more accessible without having to read the caption. Finally, the dotted lines (annually interpolated values) are hardly visible.

Figures were updated following your suggestions. We decided to remove the dotted lines.

- Figures 4 and 6: '(b and c)' should be replaced with '(b and d)'

Thank you for catching this typo. We made the corresponding corrections.

**Reply to Referee #2**

We thank referee #2 for the helpful comments. We have addressed the referee's concerns as explained below.

General comments:

- Consider adding "Subpolar" to North Atlantic in the title.

Added.

- I believe your results are occasionally strongly affected by the TTO data (particularly in the Irminger basin), conceivably worsened by your time-interpolation performed to 'provide weight to old cruises'. I recommend publication of your results with exclusion of TTO data, or least mention in the text of how such exclusion would affects results.

Thank you for your suggestion. We recalculated all the trends excluding TTO data and the difference between pH trends with and without TTO were significant. Hence, we decided to exclude the TTO data from our study, with all the changes that it entails.

- The sections plots of Fig1 and Fig2 are unrealistically noisy. The captions suggests the "mean distributions" are plotted, but these are not averages, but rather all data of all cruises thrown into a single section, with inappropriately short influence radii for the contouring (or whatever the equivalent terminology is for DIVA gridding). They thus represent not natural spatial heterogeneity, but temporal aliasing. This leads to disturbingly jittery artifacts (particularly evident in Fig2d as blue/purple/pink patchwork). Consider either contouring true averages, or simply increasing the influence radii (i.e., smooth it more).

Thank you for your suggestions. You are right, what we did is plotting the data of all cruises in a single section, rather than averaging the data. Since the purpose of describing Figure 2 is giving a general view of the properties along this section, we think that describing a single cruise is enough. For this reason, we decided to describe the general distribution of the main variables along the section using the 2004 cruise as reference. This cruise represents the mean year of the studied period. In this way we avoid interpolating the data from all cruises to a single grid and then averaging them to build Figure 2, with all the errors that this would entail.

- Consider adding a visually catchy and informative summarizing section plot (one for pHobs, or perhaps one per pH-driver), showing per water mass the rate of pH change. In each, surface layers would most red, as would DSOW, with intermediate layers slightly lower, and Iceland on average lower than Irminger.

Thank you for your suggestion. We added the suggested pHobs figure to the supplementary information (Figure S1).

Specific comments:

- Consider capturing some more cruise details in your Table 1. For instance, please tabulate the type of measurements performed on each cruise (which had pH directly, which calculated it – i.e., you lines ~75–100). What is the consequence of the rather seriously sounding, but nonchalantly made remark in line 91 "However, Carter et al reported a pH inaccuracy of 0.0055"? Is that a positive or negative bias? Systematic for everyone or just for them? Do you compensate?

We added an extra column that specifies the measurements performed on each cruise. What we wanted to highlight with the sentence in line 91 is the fact that although it is possible to achieve high reproducibility in pH measurements, all the measurements will have an inherent uncertainty of 0.0055 due to the uncertainty in the determination of the constants of the tris-buffer. Therefore, the 0.0055 quantity is an uncertainty that affects all pH measurements, which we cannot compensate. We have clarified this point by changing the highlighted sentence to: "However, Carter et al. (2013) reported an inherent uncertainty of spectrophotometric pH determinations of 0.0055 pH units, associated to the trisbuffer used for calibration". (See also answer to comment about line 131).

- line 113: less => minus

Corrected

– line 120: "advantages" relative to what?  $\Delta C^*$ ?

Yes. We added the following statement at the end of the sentence: "relative to the previous method proposed by Gruber et al. (1996)".

- line 121: I can't follow. The suggestion is that no Cant-free reference waters are required, but it's not clear why that is. Consider explaining more clearly or not at all and only keeping the reference to VR2012).

We changed the statement to make it clearer: "And second, the parameterizations of  $A_T^0$  and  $\Delta C_{diseq}$  are determined using the subsurface layer as reference (Vázquez-Rodríguez et al., 2012a), where the age of the water parcel and, therefore, its  $C_{ant}$  concentration is estimated using CFC measurements (Waugh et al., 2006)".

- line 131: explain why you consider 0.0055 the "accuracy" of the pH measurements. Again, if Carter thinks this is a /systematic/ error of the method, that would not affect detectability of trends.

You are right; this uncertainty will affect all pH measurements and, therefore, will not affect the detectability of trends. What we wanted to highlight with this statement is that the data we are using have high reproducibility (higher than the accuracy of the measurements) and thus are suitable for determining trends. We added the following statement to the manuscript: "The high reproducibility, an order of magnitude better than the uncertainty (0.0055 pH units, Carter et al. (2013)), is suggestive of high quality data". (See also answer to comment about line 91).

- I find the use of statistical terminology confusing. The terms "standard deviation", "confidence interval" and seem to be used loosely or even interchangeably while they each have a clearly defined use case. (Line 99-100 seems to suggest that you equate "two standard deviations" to "confidence interval"). If the use of these terms is nonetheless is correct then certainly the employed confidence level should be mentioned to make sense of the stated confidence intervals. I particularly object to referring to the standard deviation of depths in a defined depth (or density) range as the 'confidence interval' of depths (first column of T1).

Thank you for your comment; we noticed that we were using the term 'confidence interval' incorrectly. In lines 99-100 the term 'confidence interval' was not correctly used and, therefore, it was deleted. In the revised manuscript we changed the term 'confidence interval' with 'standard deviation'. We, therefore, change the numbers presented in the tables accordingly. We also eliminated the standard deviation of the pressure data in Table 2 as you suggested. We only used the confidence intervals in figures 3-6, where we defined what we consider as confidence interval  $(2x(standard deviation)/\sqrt{N}, where N is the number of samples), which corresponds to a 95% confidence interval since our samples are independent.$

– I hold the whole of line 125-140 to constitute a slight misuse of statistical numbers. The reasoning here seems to be "the spread between the means of cruises is smaller than the spread within each cruise, and thus we believe we can detect trends between cruises". Although the closeness in cruise means is certainly comforting, that alone does not make for detectability of trends. It would at best provide a lower bound for the detectability of trends (i.e., trends within the ranges given in T1 would go unnoticed but might nonetheless exist). Consider adding a small statement that indicates these results are suggestive of high quality, and try to avoid suggesting to provide evidence thereof.

Thank you for your suggestions. We agree with your comment. We changed lines 139-140 as follows: "The high reproducibility, an order of magnitude better than the uncertainty (0.0055 pH units, Carter et al. (2013)), is suggestive of high quality data. Using these standard deviations for the seven cruises, and taking into account the 25 years considered in this study, the threshold of detectability of pH trends at 95% of confidence is 0.00012 pH units·yr-1, which renders confidence to the estimated trends".

- line 150: this 'replacement' process is a little rash. I can imagine ignoring these shallow data, but simply overwriting them with data that has less sensitivity to seasonality without providing a compelling case for doing so is not warranted. I do not believe that ignoring the 100m surface layer would yield a vastly different result to what you now got. If that is indeed so, I recommend using that ignore-approach, to avoid the suggesting that you're fudging.

Thank you for your suggestion. We followed it by removing the photic layer (pressure

We added the following statement to clarify this question in the manuscript: "Given that the variability of the physicochemical properties within each layer is relatively low (see standard deviations of the averaged values in Table S1), we can assume that these derivatives are constant over the studied time period and use a constant derivative value for each layer".

- Consider restructuring 2.3 into a distinct paragraph for the determination of time trends and one for inferring strength of individual drivers. Your TABLE3 mentions the "sum of drivers" or "model", which terminology is nowhere used in the text, please harmonize. Also, T3 separates influence of Cnat and Cant, but Eq2 does not.

We included a table in the Supplementary Information that shows the strength of individual drivers (Table S2). Since time trends are represented in figures 4-6, rebuilding section 2.3 would involve citing figures 4-6 before figure 2, which we think this is not a logical order for figures in the manuscript. However, we added the following statement to the results section to clarify where the reader can find the derivative values: "The values of each term of  $\frac{\partial pH_{Tis}}{\partial var}$  and  $\frac{dvar}{dt}$  (where var refers to each of the drivers) described in Sect. 2.2 can be found in the Supplementary Table S2 and in Figs. 4-6, respectively".

We changed dpH/dt\_model by dpH/dt\_total, and introduced this new term to Eq. (2) and Table 3. We also added some explanatory comments throughout the manuscript to distinguish more clearly between dpH/dt\_model and dpH/dt\_total.

We slightly changed Eq. (2) to clearly indicate the separation between  $C_{ant}$  and  $C_{nat}$ :  $\left(\frac{dpH_{Tis}}{dt}\right)_{total} = \frac{\partial pH_{Tis}}{\partial T_{is}}\frac{dT_{is}}{dt} + \frac{\partial pH_{Tis}}{\partial S}\frac{dS}{dt} + \frac{\partial pH_{Tis}}{\partial A_{T}}\frac{dA_{T}}{dt} + \frac{\partial pH_{Tis}}{\partial DIC}\left(\frac{dC_{ant}}{dt} + \frac{dC_{nat}}{dt}\right)$

- line 168: "real average value" => "observed linear trend" (???)

What we meant is the average value calculated for each layer and cruise. We changed the sentence to the following to make this clearer: "To estimate  $\frac{\partial pH_{Tis}}{\partial var}$  (where var refers to each of the drivers:  $T_{is}$ , S,  $A_T$  and DIC) we calculated a  $pH_{Tis}$  for each layer and year using the layer average value of var for each year but keeping the values of the other drivers constant and equal to the time-average value for the layer over the studied time period".

- TABLE1: I believe "confidence interval" here is "standard deviation", or is it truly CI? Then state the confidence level. I'm not sure the CI of the average of averages, or however one would call the last row, has any statistical meaning – why not simply provide SD in that row? You use pH25 in table 1, while stating in the text that pHisT and pH25 are not easily compared – why the sudden use of pH25 here?

We believe you are referring to Table 2. We changed 'confidence interval' by 'standard deviation' and we changed the results accordingly. The purpose of this table is to give an estimate of the reproducibility of the analysis and calculation methods, which we take as an indicative of the goodness of the data for trend analysis. Regarding the scale of pH, since trends are performed using pH at in situ conditions, we followed your suggestion and we changed the pH reported in Table 2 by pH at in situ conditions.

- FIGURE1: some of the contour intervals have at sig1 or sig2 label, while caption and text suggest cutoffs were based on sig0.

Sorry for the confusion, sig $\theta$  would refer to all the sigma levels, either referenced to 0, 1000 or 2000 dbar. To make this clearer we changed the figure caption by "(referenced to 0 dbar,  $\sigma_0$ ; 1000 dbar,  $\sigma_1$ ; and 2000 dbar,  $\sigma_2$ ; all in kg·m-3)"

- I generally very much like your other (time-tested) figures. Perhaps increase coverage of fig 1a to provide a view of distance to land on eastern extent of section.

We updated Figure 1a following your suggestions. We increased the eastward extension of the map.

– Consider moving 3.1 to the introduction.

We appreciate your suggestion. However, and despite the qualitative nature of this section, we think it fits better in the results section rather than in the introduction, since it is a description based on our data. Besides, it provides useful information for understanding the sections coming next. In view of this, we decided to keep this section as it is.

– line 209: "almost homogenous" – sections plots suggest otherwise, see earlier comment on influence radii

After changing Figure 2 as suggested in the previous comment regarding this figure, now the statement is true.

- line 211: "because they are correlated" – that relationship is not causal, please rephrase.

You are right. We eliminated this statement, so that, in our opinion, the explanation of property distributions is improved.

– line 212: can you qualify that "80%" in light of the mentioned  $\Delta$ Cdiseq? Is this what one would expect?

The existence of a saturation lower than 100% in the surface layer is expectable for an oceanic uptaken of  $C_{ant}$  (see Matsumoto and Gruber, 2006). Therefore, a disequilibrium in the  $C_{ant}$  content with respect to the atmospheric  $C_{ant}$  is expected.

Matsumoto, K. and Gruber, N.: How accurate is the estimation of anthropogenic carbon in the ocean? An evaluation of the  $\Delta C^*$  method, Global Biogeochem. Cycles, 19, doi:10.1029/2004GB002397, 2005.

– line 239: can you speculate on the possible causes for the supposedly spuriously high rate of pH decrease observed by Bates et al at IrmSTS?

We added the explanation given by Bates et al. (2014): "Bates et al. (2014) linked the high acidification rate found at the Irminger Sea time-series to the high rate of increase in DIC ( $1.62 \pm 0.35 \ \mu mol \cdot kg^{-1} \cdot yr^{-1}$ ) observed at this site, which is almost twice our rate of increase in DIC ( $0.64 \pm 0.07 \ \mu mol \cdot kg^{-1} \cdot yr^{-1}$ , Fig. 5c). This is based on data from only one site, further north than our section, and indicates that spatial variations are substantial in this region". We do not have enough information to elucidate other possible explanations.

- line 244: you mention a tropical Pacific time series station, and contrast it with your work and a subpolar Pacific TSS, latter two match nicely. Add a brief sentence attributing that contrast.

Now that we are not including the TTO cruise data in our study, our trends are in agreement with the trends found in the tropical Pacific and are slightly higher than the trends found in the subpolar Pacific. We changed the manuscript accordingly to these changes, and we added the following explanation to the low trends found in the subpolar Pacific: "Wakita et al. (2013) attributed the lower than expected pH trends to an increasing  $A_T$  trend".

- line 246-254: your statement "renders direct comparison difficult" does not stand up to scrutiny. Recalculating pH to different temperatures does not changes the slope of a pH trend. That is, slopes can be compared (i.e., VR12's Fig3ab vs your Fig3ab), even if absolute values cannot. I recommend more work is made of this comparison, particularly if results between studies differ.

The fact that precludes comparing trends is not the difference in the temperature at which pH is referenced, but the fact that Vazquez-Rodriguez et al. (2012b) normalized pH values to potential temperature, salinity, silicate and AOU in WOA05. This normalization eliminates part of the influence of these parameters (potential temperature, salinity, silicate and AOU) on the pH trends. This is why direct comparison between their pH trends and our pH trends is difficult. However, we added some text comparing the trends reported by both works: "This normalization, combined with the different temporal coverage (1981–2008), causes the rates reported by Vazquez-Rodriguez et al. (2012b) differ from those obtained in the present work. The pHN trends reported for the SPMW and uLSW layers of the Irminger basin and for the ISOW layer of the Iceland basin are very similar to our pHTis trends for these layers. However, the pHN trends reported by Vazquez-Rodriguez et al. (2012b) for the cLSW layer in both basins and for the ISOW layer in the Irminger basin are significantly different from our pHTis trends for these layers, but are very similar to pH changes derived from Cant changes  $\left(\frac{\partial pH_{Tis}}{\partial DIC}\frac{dC_{ant}}{dt}\right)$  in Table 3). In the case of the DSOW layer, the pHN trend is also in agreement with  $\frac{\partial pH_{Tis}}{\partial DIC}\frac{dC_{ant}}{dt}$  trends. This suggests that the normalization carried out by Vazquez-Rodriguez et al. (2012b) could remove some of the impact of the natural component (represented here by  $C_{nat}$ ) over pH changes, essentially due to the use of AOU in the normalization".

- line 322: if anything, these are 4 decades (80s 90s 00s and 10s). Consider "34-year period" or similar

We changed it by "25-year period", since now we are not considering the TTO cruise.

- line 323 (and likely elsewhere): "separate and increase into its drivers" is slightly sloppy English. Consider rephrasing

We changed the sentence into: "From the study of the main drivers of the observed pH changes...".

- line 325: "However" => "thus". Reduced rate of decrease of pH is what one expects with increasing alk.

Changed.

- line 327: "salty" => "saline"

Changed.

- line 333: consider "observe" => "infer". There's too many interpretative steps involved to call this "observe"

Changed.

Author's changes in manuscript: List of major changes in the revised manuscript

**List of major changes in the revised manuscript**

- 1) The text has been revised according to all the referees' comments. In particular, the methodology section has been significantly revised to make clearer the procedure followed in our work to achieve the results presented.
- 2) Title has been slightly modified to fulfill the referees' suggestions.
- 3) Data from the older cruise has been excluded from our study.
- 4) Surface data (first 75 dbar) has not been included.
- 5) Comparison with previous works has been extended.
- 6) Map on Fig. 1 has been color updated and eastward extended.
- 7) Figure 2 has been updated and now shows data from one cruise instead of plotting the data of all cruises in a single section plot.
- 8) Figures 3-6 have been updated to make them more readable on black and white version, and titles of the basins have been added on top of the figures.

1 2

**Ocean acidification in the Subpolar North Atlantic: rates and mechanisms controlling mechanismspH changes**

Maribel I. García-Ibáñez1, Patricia Zunino2, Friederike Fröb3, Lidia I. Carracedo4, Aida F.
 Ríos†, Herlé Mercier5, Are Olsen3, Fiz F. Pérez1

[revised manuscript text omitted]

197To estimate how much each of these altogether five factors have
assumed linearity and decomposed the observed pH changes into these potential drivers according to:

| 100 | арнтіз _ д              | pH Tis dT is _         | <del>∂pHTis dS</del> ⊥                         | <del>∂pHTis dAT</del> ⊥             | $\frac{\partial p H_{Tis}}{d(C_{nat}+C_{ant})}$                                                                                                                                    |                                                     | (2) |
|-----|-------------------------------------------|----------------------------------------------|-----------------------------------------------------------|-----------------------------------------------------------|------------------------------------------------------------------------------------------------------------------------------------------------------------------------------------|-----------------------------------------------------|-----|
| 177 | <del>dt</del> -                           | ∂T is dt '                        | ∂s dt '                                                   | ∂A ∓ dt ′                                      | <del>dDIC</del> dt '                                                                                                                                                               |                                                     | (2) |
| 200 | $\left(\frac{dpH_{Tis}}{dt}\right)_{tot}$ | $\frac{\partial p H_{Tis}}{\partial T_{is}}$ | $\frac{dT_{is}}{dt} + \frac{\partial pH_{T}}{\partial S}$ | $\frac{dS}{dt} + \frac{\partial p H_{Tis}}{\partial A_T}$ | $\frac{\mathrm{dA}_{\mathrm{T}}}{\mathrm{dt}} + \frac{\partial \mathrm{pH}_{\mathrm{Tis}}}{\partial \mathrm{DIC}} \left( \frac{\mathrm{dC}_{\mathrm{ant}}}{\mathrm{dt}} + \right.$ | $\frac{\mathrm{d}C_{\mathrm{nat}}}{\mathrm{d}t}$ ), | (2) |

Due to the small range of pH change to which we are working and to the relatively low pH variability within
 each layer, we can consider that pH follows a linear scale instead of a logarithmic scale. This causes that the
 contributions of each of the terms considered in Eq. (2) to pH change are equivalent to the contributions in terms
 of [H+].

To estimate  $\frac{\partial p H_{Tis}}{\partial war}$  (where *var* refers to each of the drivers: Tis, S, AT and DIC) we calculated the meana pHTis 205 206 for each layer and eruiseyear using the reallayer 
[revised manuscript text omitted]

- $\frac{A_{T} \text{ from dissolution of CaCO}_{3}. \text{ The influence of these high } A_{T} \text{ values is then transported by the ISOW}$   $\frac{A_{T} \text{ from dissolution of CaCO}_{3}. \text{ The influence of these high } A_{T} \text{ values is then transported by the ISOW}$   $\frac{A_{T} \text{ from dissolution of CaCO}_{3}. \text{ The influence of these high } A_{T} \text{ values is then transported by the ISOW}$
- 271 **3.2** Water mass acidification and drivers

272 Trends of pHTis in each layer and basin are presented in Table 3-and, in Fig. 3 and in Supplementary Fig. S1. 273 The pHTis has decreased in all layers of the Irminger and Iceland basins during the time period of more than  $\frac{320}{2}$ 274 years (19891–2015) that is covered by our the data. The trends are stronger in the Irminger basin due to the 275 presence of younger waters. The rate of OApH decline decreases with depth, except for the DSOW layer that has 276 acidification rates close to those found in the cLSW layer. This indicates that DSOW is a newly formed water 277 mass that has recently been in contact with the atmosphere. Moreover, the acidification rate in the ISOW layer in 278 the Irminger basin is relatively low, which could be related to the increasing importance on this layer of the 279 relatively old NADW in this layer, with the diminutionreduction in-volume of cLSW formation since mid-90s 280 (Lazier et al., 2002; Yashayaev, 2007).

The observed rate of  $pH_{Tis}$  decrease in the SPMW layer of the Iceland basin (-0.00126 ± 0.0001 pH units  $\cdot yr^{-1}$ ; 281 282 Table 3, Fig. 3b) is in agreement with that observed at the Iceland Sea time-series (68°N, 12.66°W; Olafsson et al. (2009, 2010)) for the period 1983–2014 (-0.0014  $\pm$  0.0005 pH units-Bates et al. (2014)). Our rates in the 283 SPMW layer of both basins are slightly lower than those observed at the Subtropical Atlantic time series stations 284 ESTOC (29.04°N, 15.50°W; Santana Casiano et al. (2007). González Dávila et al. (2010)) for the period 1995 285 2014 (0.0018 ± 0.0002 pH units yr+; Bates et al. (2014)) and BATS (32°N, 64°W; Bates et al. (2014)) for the 286 period 1983 2014 (  $0.0017 \pm 0.0001$  pH units yr+; Bates et al. (2014)). However, our rate of pHTis decrease in 287 the SPMW layer in the Irminger basin  $(-0.001\frac{38}{2} \pm 0.0001 \text{ pH units} \cdot \text{yr}^{-1})$  is only half of lower than that observed 288 289 in the sea surface waters of the Irminger Sea time-series (64.3°N, 28°W; Olafsson et al. (2010)) for the period 1983–2014 (-0.0026  $\pm$  0.0006 pH units-vr-1; Bates et al. (2014)), which is exceptionally high compared to the 290 291 other time series summarized here. Bates et al. (2014) linked the high acidification rate found at the Irminger Sea 292 time-series to the high rate of increase in DIC  $(1.62 \pm 0.35 \,\mu\text{mol}\cdot\text{kg}^{-1}\cdot\text{yr}^{-1})$  observed at this site, which is almost twice our rate of increase in DIC ( $0.64 \pm 0.07 \mu \text{mol} \cdot \text{kg}^{-1} \cdot \text{yr}^{-1}$ , Fig. 5c). This is based on data from only one site, 293 294 further north than our section, and indicates that spatial variations are substantial in this region. Besides, the 295 acidification rates in the SPMW layer of the both basins here reported are in agreement with the rates of -0.0020  $\pm$  0.0004 pH units yr-1 determined for the North Atlantic subpolar seasonally stratified biome for the period 296 297 1991-2011 (Lauvset et al., 2015). Compared to the Subtropical Atlantic time-series stations, oour rates in the 298 SPMW layer of both basins are slightlyin agreement with-lower than those observed at the Subtropical Atlantic time-series stations-ESTOC (29.04°N, 15.50°W; Santana-Casiano et al. (2007), González-Dávila et al. (2010)) 299 for the period 1995–2014 (-0.0018 ± 0.0002 pH units yr-1; Bates et al. (2014)) and BATS (32°N, 64°W; Bates et al. 300 al. (2014)) for the period 1983–2014 (-0.0017  $\pm$  0.0001 pH units yr-1; Bates et al. (2014)). Comparinged to with 301 302 the Pacific Ocean, the OA rates in the Iceland and Irminger basins are slightly lower thanin agreement with those 303 reported for the Central North Pacific based on data from the time-series station HOT (22.45°N, 158°W; Dore et al. (2009)) for the period 1988–2014 (-0.0016  $\pm$  0.0001 pH units vr-1; Bates et al. (2014)), but are in agreement 304 305 withslightly higher than those founddetermined by Wakita et al. (2013) in the winter mixed layer at the Subarctic 306 Western North Pacific (time-series stations K2 and KNOT) for the period 1997–2011 (-0.0010  $\pm$  0.0004 pH 307 units  $vr^{-1}$ ). Wakita et al. (2013) attributed the lower than expected pH trends to an increasing  $A_T$  trend.

[revised manuscript text omitted]

348of the Iceland basin, which can be explained by the circulation and mixing of this layer. As ISOW flows349downstream along the Reykjanes Ridge, it mixes with cLSW and NADW (van Aken and de Boer, 1995;350Fogelqvist et al., 2003). The reduced volume of cLSW since mid 90s (Lazier et al., 2002; Yashayaev, 2007) has351increased the importance of NADW (with high  $A_1$ ; Fig. 2h) in the ISOW layer, making the pH decrease of the352ISOW layer of the Iceland basin lower than in the Irminger basin.

353 The DIC increase (Fig. 5c,d) is the main cause of the observed pH decreases (Table 3), and corresponds to pH drops between -0.0008599 and -0.00134205 pH units yr-1 (Table 3). The waters in both the Irminger and Iceland 354 355 basins gained DIC in response to the increase in atmospheric CO2; the convection processes occurring in these 356 basinsthe former basin (Pickart et al., 2003; Thierry et al., 2008; de Boisséson et al., 2010; García-Ibáñez et al., 357 2015; Fröb et al., 2016; Piron et al., 2016) and in the surrounding ones (i.e., Labrador and Nordic Seas) provide 358 an important pathway for DIC to pass from the surface mixed layer to the intermediate and deep layers. The 359 effect of the DIC increase on pH is generally dominated by the anthropogenic component (Table 3). The 360 exception comes with the cLSW layer of the Irminger basin, where dominates the natural component resulting 361 from the aging of the layer dominates. In general, the Irminger basin All layers have higher Cant increase rates in 362 the Irminger basin than in the Iceland basin layers (Fig. 6a,b), and therefore larger pH declines, presumably a 363 result of the proximity convection in the Irminger basin itself and advection of newly ventilated waters from the Irminger basin to the regions of deep water formationLabrador Sea. The highest Cant increase rates are found in 364 365 the SPMW layer, owing to its direct contact with the atmosphere, and result in the highest strongest rates of pH 366 decrease. The higher pH drops related to Cinit increase found in the SPMW layer in the Irminger basin compared to those found in the Iceland basin layer, can be related to the differences in the rise in Cant levels in both basins. 367 368 In the Irminger basin, the rise in  $C_{ant}$  levels of the SPMW layer correspond to about 857% of the rate expected 369 from a surface ocean maintaining its degree of saturation with the atmospheric  $CO_2$  rise (computed using as 370 reference the measurements of Mauna Loa)globally averaged marine surface annual mean pCO2 data from the 371 NOAA, ftp://aftp.cmdl.noaa.gov/products/trends/co2/co2 annmean gl.txt), while in the Iceland basin, this rate is 372 about 73% of the expected rate. The lower fraction